# Neural Link Prediction with Walk Pooling

**Liming Pan**[*]
School of Computer and Electronic Information,
Nanjing Normal University,
210023 Nanjing, China.
`pan.liming@njnu.edu.cn`

**Cheng Shi**[*] **& Ivan Dokmanić** [†]
Departement Mathematik und Informatik,
Universität Basel,
4051 Basel, Switzerland.
`{firstname.lastname}@unibas.ch`

## Abstract

Graph neural networks achieve high accuracy in link prediction by jointly leveraging graph topology and node attributes. Topology, however, is represented indirectly; state-of-the-art methods based on subgraph classification label nodes with distance to the target link, so that, although topological information is present, it is tempered by pooling. This makes it challenging to leverage features like loops and motifs associated with network formation mechanisms. We propose a link prediction algorithm based on a new pooling scheme called WalkPool. WalkPool combines the expressivity of topological heuristics with the feature-learning ability of neural networks. It summarizes a putative link by random walk probabilities of adjacent paths. Instead of extracting transition probabilities from the original graph, it computes the transition matrix of a "predictive" latent graph by applying attention to learned features; this may be interpreted as feature-sensitive topology fingerprinting. WalkPool can leverage unsupervised node features or be combined with GNNs and trained end-to-end. It outperforms state-of-the-art methods on all common link prediction benchmarks, both homophilic and heterophilic, with and without node attributes. Applying WalkPool to a set of unsupervised GNNs significantly improves prediction accuracy, suggesting that it may be used as a general-purpose graph pooling scheme.

## 1 Introduction

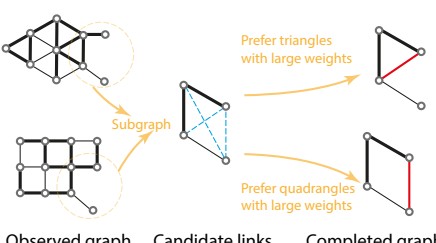

Figure 1: The topological organizing rules are not universal across graphs.

Graphs are a natural model for relational data such as coauthorship networks or the human protein interactome. Since real-world graphs are often only partially observed, a central problem across all scientific domains is to predict missing links (Liben-Nowell & Kleinberg, 2007). Link prediction finds applications in predicting protein interactions (Qi et al., 2006), drug responses (Stanfield et al., 2017), or completing the knowledge graph (Nickel et al., 2015). It underpins recommender systems in social networks (Adamic & Adar, 2003) and online marketplaces (Lü et al., 2012). Successful link prediction demands an understanding of the principles behind graph formation.

In this paper we propose a link prediction algorithm which builds on two distinct traditions: (1) complex networks, from which we borrow ideas about the importance of topology, and (2) the emerging domain of graph neural networks (GNNs), on which we rely to learn optimal features. We are motivated by the fact that the existing GNN-based link prediction algorithms encode topological[1] features only indirectly, while link prediction is a strongly

---

[*]These two authors have equal contribution.

[†]To whom correspondence should be addressed.

[1]Mathematical topology studies (global) properties of shapes that are preserved under homeomorphisms. Our use of "topology" to refer to local patterns is common in the network literature.

topological task (Lü & Zhou, 2011). Traditional heuristics assume certain topological organizing rules, such as preference for triangles or quadrangles; see Figure 1 for an illustration. But organizing rules are not universal across graphs and they need to be learned. The centerpiece of our algorithm is a new random-walk-based pooling mechanism called WALKPOOL which may be interpreted as a learnable version of topological heuristics. Our algorithm is universal in the sense of Chien et al. (2020) in that it models both heterophilic (e.g., 2D-grid in Figure 1) and homophilic (e.g., Triangle lattice in Figure 1) graphs unlike hardcoded heuristics or standard GNNs which often work better on homophilic graphs. WALKPOOL can learn the topological organizing rules in real graphs (for further discussion of lattices from Figure 1 see also Section 4.4), and achieves state-of-the-art link prediction results on all common benchmark datasets, sometimes by a significant margin, even on datasets like **USAir** (Batagelj & Mrvar, 2006) where the previous state of the art is as high as 97%.

## 1.1 RELATED WORK

Early studies on link prediction use heuristics from social network analysis (Lü & Zhou, 2011). The homophily mechanism for example (McPherson et al., 2001) asserts that "similar" nodes connect. Most heuristics are based on connectivity: the common-neighbor index (CN) scores a pair of nodes by the number of shared neighbors, yielding predicted graphs with many triangles; Adamić–Adar index (AA) assumes that a highly-connected neighbor contributes less to the score of a focal link (Adamic & Adar, 2003). CN and AA rely on paths of length two. Others heuristics use longer paths, explicitly accounting for long-range correlations (Lü & Zhou, 2011). The Katz index (Katz, 1953) scores a pair of nodes by a weighted sum of the number of walks of all possible lengths. PageRank scores a link by the probability that a walker starting at one end of the link reaches the other in the stationary state under a random walk model with restarts.

Heuristics make strong assumptions such as a particular parametric decay of path weights with length, often tailored to either homophilic or heterophilic graphs. Moreover, they cannot be easily used on graphs with node or edge features. Instead of hard-coded structural features, link prediction algorithms based on GNNs like the VGAE (Kipf & Welling, 2016b) or SEAL Zhang & Chen (2018) use learned node-level representations that capture both the structure and the features, yielding unprecedented accuracy.

Two strategies have proved successful for GNN-based link prediction. The first is to devise a score function which only depends on the two nodes that define a link. The second is to extract a subgraph around the focal link and solve link prediction by (sub)graph classification (Zhang & Chen, 2017). In both cases the node representations are either learned in an unsupervised way (Kipf & Welling, 2016b; Mavromatis & Karypis, 2020) or computed by a GNN trained joinly with the link classifier (Zhang & Chen, 2018; Zhang et al., 2020).

Algorithms based on two-body score functions achieved state-of-the-art performance when they appeared, but real networks brim with many-body correlations. Graphs in biochemistry, neurobiology, ecology, and engineering (Milo et al., 2004) exhibit *motifs*—distinct patterns occuring much more often than in random graphs. Motifs may correspond to basic computational elements and relate to function (Shen-Orr et al., 2002). A multilayer GNN generates node representation by aggregating information from neighbors, capturing to some extent many-body and long-range correlations. This dependence is however indirect and complex topological properties such as frequency of motifs are smoothed out.

The current state-of-the-art link prediction algorithm, SEAL (Zhang & Chen, 2018), is based on subgraph classification and thus may account for higher-order correlations. Unlike in vanilla graph classification where a priori all links are equaly important, in graph classification for link prediction the focal link plays a special role and the relative placement of other links with respect to it matters. SEAL takes this into account by labeling each node of the subgraph by its distance to the focal link. This endows the GNN with first-order topological information and helps distinguish relevant nodes in the pooled representation. Zhang and Chen show that node labeling is essential for SEAL's exceptional accuracy. Nevertheless, important structural motifs are represented indirectly (if at all) by such labeling schemes, even less so after average pooling. We hypothesize that a major bottleneck in link prediction comes from suboptimal pooling which fails to account for topology.

## 1.2 Our contribution

Following Zhang & Chen (2018), we approach link prediction from via subgraph classification. Instead of encoding relative topological information through node labels, we design a new pooling mechanism called WALKPOOL. WALKPOOL extracts higher-order structural information by encoding node representations and graph topology into random-walk transition probabilities on some effective *latent* graph, and then using those probabilities to compute features we call *walk profiles*. WALKPOOL generalizes loop-counting ideas used to build expressive graph models (Pan et al., 2016). Computing expressive topological descriptors which are simultaneously sensitive to node features is the key difference between WALKPOOL and SEAL. Using normalized probabilities tuned by graph attention mitigates the well-known issue in graph learning that highly-connected nodes may bias predictions. Transition probabilities and the derived walk profiles have simple topological interpretations.

WALKPOOL can be applied to node representations generated by any unsupervised graph representation models or combined with GNNs and trained in end-to-end. It achieves state-of-the-art results on all common link prediction benchmarks. Our code and data are available online at `https://github.com/DaDaCheng/WalkPooling`.

## 2 Link prediction on graphs

We consider an *observed* graph with $N$ nodes (or vertices), $\mathcal{G}^o = (\mathcal{V}, \mathcal{E}^o)$, with $\mathcal{V}$ being the node set and $\mathcal{E}^o$ the set of *observed* links (or edges). The observed link set $\mathcal{E}^o$ is a subset $\mathcal{E}^o \subset \mathcal{E}^*$ of the set of all true links $\mathcal{E}^*$. The target of link prediction is to infer missing links from a candidate set $\mathcal{E}^c$, which contains both true (in $\mathcal{E}^*$) and false (not in $\mathcal{E}^*$) missing links.

**Problem 1.** *(Link prediction) Design an algorithm* `LearnLP` *that takes an observed graph* $\mathcal{G}^o \subset \mathcal{G}$ *and produces a link predictor* `LearnLP`$(\mathcal{G}^o) = \Pi$ $: \mathcal{V} \times \mathcal{V} \to \{\text{True}, \text{False}\}$ *which accurately classifies links in* $\mathcal{E}^c$.

Well-performing solutions to Problem 1 exploit structural and functional regularities in the observed graph to make inferences about unobserved links.

**Path counts and random walks** For simplicity we identify the $N$ vertices with integers $1, \ldots, N$ and represent $\mathcal{G}$ by its adjacency matrix, $\mathbf{A} = (a_{ij})_{i,j=1}^N \in \{0,1\}^{N \times N}$ with $a_{ij} = 1$ if $\{i, j\} \in \mathcal{E}^o$ and $a_{ij} = 0$ otherwise. Nodes may have associated feature vectors ($\mathbf{x}_i \in \mathbb{R}^F, i \in \{1, \ldots, N\}$); we collect all feature vectors in the feature matrix $\mathbf{X} = [\mathbf{x}_1, \cdots, \mathbf{x}_N]^T \in \mathbb{R}^{N \times F}$.

Integer powers of the adjacency matrix reveal structural information: $[\mathbf{A}^\tau]_{ij}$ is the number of paths of length $\tau$ connecting nodes $i$ and $j$. WALKPOOL relies on random walk transition matrices. For an adjacency matrix $\mathbf{A}$ the transition matrix is defined as $\mathbf{P} = \mathbf{D}^{-1}\mathbf{A}$ where $\mathbf{D} = \text{diag}(d_1, \ldots, d_N)$ and $d_i = \sum_j a_{ij} = |\mathcal{N}(i)|$ is the degree of the node $i$. Thus the probability $p_{ij} = d_i^{-1}a_{ij}$ that a random walker at node $i$ transitions to node $j$ is inversely proportional to the number of neighbors of $i$. An extension to graphs with non-negative edge weights $\mathbf{W} = (w_{ij})$ is straightforward by replacing $a_{ij}$ by $w_{ij}$. Powers of $\mathbf{P}$ are interpretable: $[\mathbf{P}^\tau]_{ij}$ is the probability that a random walker starting at node $i$ will reach node $j$ in $\tau$ hops.

As we show in Section 3.2, transition probabilities in WALKPOOL are determined as coefficients of an attention mechanism applied to learned node features. The node features are in turn extracted by a parameterized function $f_\theta$, which distills input attributes and to an extent graph topology into an embedding (a vector) for each node. The feature extractor $f_\theta : \{0,1\}^{N \times N} \times \mathbb{R}^{N \times F} \to \mathbb{R}^{N \times F'}$ takes as input the adjacency matrix (which encodes the graph topology) and the input feature matrix, and outputs a distilled node feature matrix. It should thus be equivariant to node ordering in $\mathbf{A}$ which is satisfied by GNNs.

## 3 WalkPool for link prediction by subgraph classification

We now describe WALKPOOL which directly leverages higher-order topological correlations without resorting to link labeling schemes and without making strong a priori assumptions. WALKPOOL first samples the $k$-hop subgraph $\mathcal{G}^k_{\{i,j\}} \subset \mathcal{G}^o$ enclosing the target link; and then computes random-walk profiles for sampled subgraphs with and without the target

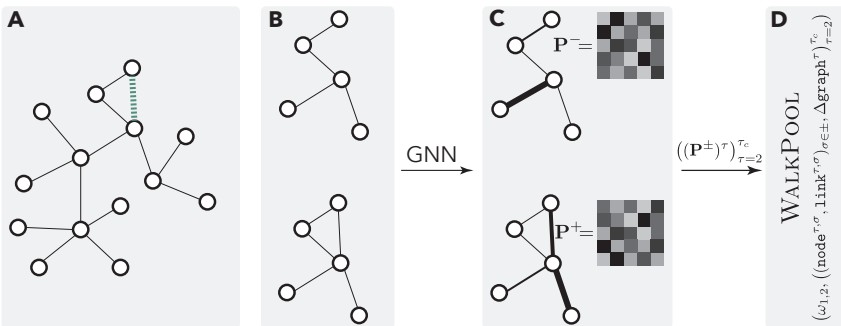

Figure 2: Illustration of WALKPOOL. **A**: The input graph and the focal link $e$; **B**: enclosing subgraphs with and without $e$; **C**: attention-processed features $\equiv$ random walk transition probabilities; **D**: extracted walk profile.

link. Random walk profiles are then fed into a link classifier. Computing walk profiles entails

1. **Feature extraction**: $\mathbf{Z} = f_\theta(\mathbf{A}, \mathbf{X})$, where $f_\theta$ is a GNN;
2. **Transition matrix computation**: $\mathbf{P} = \texttt{AttentionCoefficients}_\theta(\mathbf{Z}; \mathcal{G})$;
3. **Walk profiles**: Extract entries from $\mathbf{P}^\tau$ for $2 \leq \tau \leq \tau_c$ related to the focal link.

We emphasize that we do not use attention to compute per-node linear combinations of features like Veličković et al. (2017) (which is analogous to Vaswani et al. (2017)), but rather interpret the attention *coefficients* as random walk transition probabilities. The features $\mathbf{Z}$ may be obtained either by an unsupervised GNN such as VGAE (Kipf & Welling, 2016b), or they may be computed by a GNN which is trained jointly with WALKPOOL.

## 3.1 SAMPLING THE ENCLOSING SUBGRAPHS

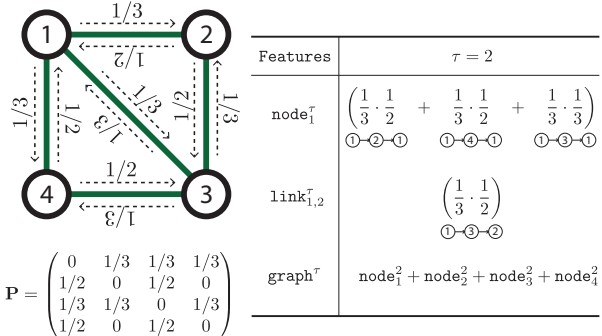

$$\mathbf{P} = \begin{pmatrix} 0 & 1/3 & 1/3 & 1/3 \\ 1/2 & 0 & 1/2 & 0 \\ 1/3 & 1/3 & 0 & 1/3 \\ 1/2 & 0 & 1/2 & 0 \end{pmatrix}$$

| Features | $\tau = 2$ |
|---|---|
| $\text{node}_1^\tau$ | $\left(\frac{1}{3}\cdot\frac{1}{2} \; + \; \frac{1}{3}\cdot\frac{1}{2} \; + \; \frac{1}{3}\cdot\frac{1}{3}\right)$ ①→②→① ①→④→① ①→③→① |
| $\text{link}_{1,2}^\tau$ | $\left(\frac{1}{3}\cdot\frac{1}{2}\right)$ ①→③→② |
| $\text{graph}^\tau$ | $\text{node}_1^2 + \text{node}_2^2 + \text{node}_3^2 + \text{node}_4^2$ |

Figure 3: Illustration of walk profiles for $\tau = 2$. We assume $p_{i,j} = 1/d_i$ where $d_i$ is the degree of node $i$.

Following earlier work we make the assumption that the presence of a link only depends on its neighbors within a (small) radius $k$. It is known that simple heuristics like AA may already perform well on many graphs, with longer walks not bringing about significant gains. Indeed, SEAL (Zhang & Chen, 2018) exploits the fact that an order $k$ heuristic can be accurately calculated from a $k$-hop subgraph; optimal $k$ is related to the underlying generative model. Keeping $k$ small (as small as 2 in this paper) is pragmatic for reasons beyond diminishing returns: a large $k$ greatly increases memory and compute demands. The size of a 2-hop neighborhood is graph-dependent, but for the highly-connected E.coli and PB datasets we already need to enforce a maximum number of nodes per hop to fit the subgraphs in memory.

Let $d(x, y)$ be the shortest-path distance between nodes $x$ and $y$. The $k$-hop enclosing subgraph $\mathcal{G}^k_{\{i,j\}}$ for $\{i, j\}$ is defined as the subgraph induced from $\mathcal{G}^o$ by the set of nodes

$$\mathcal{V}^k_{\{i,j\}} = \{x \in \mathcal{V} \; : \; d(x, i) \leq k \text{ or } d(x, j) \leq k\}.$$

Then $\mathcal{G}^k_{\{i,j\}} = (\mathcal{V}^k_{\{i,j\}}, \mathcal{E}^k_{\{i,j\}})$, where $\{x, y\} \in \mathcal{E}^k_{i,j}$ when $x, y \in \mathcal{V}^k_{\{i,j\}}$ and $\{x, y\} \in \mathcal{E}^o$. We omit the dependence on $k$ and write $G_{\{i,j\}} = (\mathcal{V}_{\{i,j\}}, \mathcal{E}_{\{i,j\}})$ for simplicity.

We fix an arbitrary order of the nodes in $\mathcal{V}_{\{i,j\}}$ and denote the corresponding adjacency matrix by $\mathbf{A}_{\{i,j\}}$. Without loss of generality, we assume that under the chosen order of

nodes, the nodes $i$ and $j$ are labeled as 1 and 2 so that the candidate link $\{i, j\}$ is mapped to $\{1, 2\}$ in its enclosing subgraph. We denote the corresponding node feature matrix $\mathbf{Z}_{\{i,j\}}$, with values inherited from the full graph (the rows of $\mathbf{Z}_{\{i,j\}}$ are a subset of rows of $\mathbf{Z}$).

For the candidate set of links $\mathcal{E}^c$, we construct a set of enclosing subgraphs $\mathcal{S}^c = \{G_{\{i,j\}} : \{i, j\} \in \mathcal{E}^c\}$, thus transforming the link prediction problem into classifying these $k$-hop enclosing subgraphs. For training, we sample a set of known true and false edges $\mathcal{E}^t$ and construct its corresponding enclosing subgraph set $\mathcal{S}^t = \{G_{i,j} : (i, j) \in \mathcal{E}^t\}$.

## 3.2 RANDOM-WALK PROFILE

The next step is to classify the sampled subgraphs from their adjacency relations $\mathbf{A}_{\{i,j\}}$ and node representations $\mathbf{Z}_{\{i,j\}}$. Inspired by the walk-based heuristics, we employ random walks to infer higher-order topological information. Namely, for a subgraph $(\mathcal{G} = (\mathcal{V}, \mathcal{E}), \mathbf{Z})$ (either in $\mathcal{S}^c$ or $\mathcal{S}^t$) we encode the node representations $\mathbf{Z}$ into edge weights and use these edge weights to compute transition probabilities of a random walk on the underlying graph. Probabilities of walks of various lengths under this model yield a *profile* of the focal link.

We first encode two-node correlations as effective edge weights,

$$\omega_{x,y} = Q_\theta(\mathbf{z}_x)^T K_\theta(\mathbf{z}_y) / \sqrt{F''}, \tag{1}$$

for all $\{x, y\} \in \mathcal{E}$, where $Q_\theta : \mathbb{R}^{F'} \to \mathbb{R}^{F''}$ and $K_\theta : \mathbb{R}^{F'} \to \mathbb{R}^{F''}$ are two multilayer perceptrons (MLPs) and $F''$ is the output dimension of the MLPs. In order to include higher-order topological information, we compute the random-walk transition matrix $\mathbf{P} = (p_{x,y})$ from the two-body correlations. We set

$$p_{x,y} = \left[\text{softmax}\left((\omega_{x,z})_{z \in \mathcal{N}(x)}\right)\right]_y := \exp(\omega_{x,y}) \, / \sum_{z \in \mathcal{N}(x)} \exp(\omega_{x,z}) \tag{2}$$

for $\{x, y\} \in \mathcal{E}$ and $p_{x,y} = 0$ otherwise, with $\mathcal{N}(x)$ the set of neighbors of $x$ in the enclosing subgraph. The encoding (2) is analogous to graph attention coefficients (Veličković et al., 2017; Shi et al., 2020); unlike graph attention, however, we directly use the coefficients instead of computing linear combinations; this framework also allows multi-head attention.

Entries of the $\tau$-th power $[\mathbf{P}^\tau]_{ij}$ are interpreted as probabilities that a random walker goes from $i$ to $j$ in $\tau$ hops. These probabilities thus concentrate the topological *and* node attributes relevant for the focal link into the form of random-walks:

- Topology is included indirectly through the GNN-extracted node features $\mathbf{Z}$, and directly by the fact that $\mathbf{P}$ encodes zero probabilities for non-neighbors and that its powers thus encode probabilities of paths and loops;
- Input features are included directly through the GNN-extracted node features, and refined and combined with topology by the key and value functions $Q_\theta$, $K_\theta$.

As a result, WALKPOOL[2] can be interpreted as trainable heuristics.

From the matrix $\mathbf{P}$ and its powers, we now read a list of features to be used in graph classification. We compute node-level, link-level, and graph-level features:

$$\texttt{node}^\tau = [\mathbf{P}^\tau]_{1,1} + [\mathbf{P}^\tau]_{2,2}, \qquad \texttt{link}^\tau = [\mathbf{P}^\tau]_{1,2} + [\mathbf{P}^\tau]_{2,1}, \qquad \texttt{graph}^\tau = \text{tr}[\mathbf{P}^\tau]. \tag{3}$$

Computation of all features is illustrated for $\tau = 2$ in Figure 3. Node-level features $\texttt{node}^\tau$ describe the loop structure around the candidate link (recall that $\{1, 2\}$ is the focal link in the subgraph). The summation ensures that the feature is invariant to the ordering of $i$ and $j$, consistent with the fact that we study undirected graphs. Link-level features $\texttt{link}^\tau$ give the symmetrical probability that a length-$\tau$ random walk goes from node 1 to 2. Finally, graph-level features $\texttt{graph}^\tau$ are related to the total probability of length-$\tau$ loops. All features depend on the node representation $\mathbf{Z}$; we omit this dependence for simplicity. The use of $\mathbf{P}$ in WALKPOOL is different from how graph matrices (e.g., $\mathbf{A}$) are used in GNNs. In GNNs, the powers $\mathbf{A}^\tau$ serve as shift operators between neighborhoods that are multiplied by filter weights and used to weigh node features; WALKPOOL encodes graph signals into effective edge weights and directly extracts topological information from the entries of $\mathbf{P}^\tau$.

[2]The output of pooling (e.g., in a CNN) is a often an object of the same type (e.g., a downsampled image). The last layer of a CNN or GNN involves global average pooling which is a graph summarization mechanism similarly as WALKPOOL, hence the name.

### 3.3 Perturbation extraction

A true link is by definition always present in its enclosing subgraph while a negative link is always absent. This leads to overfitting if we directly compute the features (3) on the enclosing subgraphs since the presence or absence of the link has a strong effect on walk profiles. For a normalized comparison of true and false links, we adopt a perturbation approach. Given an enclosing subgraph $\mathcal{G} = (\mathcal{V}, \mathcal{E})$, we define its variants $\mathcal{G}^+ = (\mathcal{V}, \mathcal{E} \cup \{1, 2\})$ (resp. $\mathcal{G}^- = (\mathcal{V}, \mathcal{E} \backslash \{1, 2\})$) with the candidate link forced to be present (resp. absent). We denote the node-level features (3) computed on $\mathcal{G}^+$ and $\mathcal{G}^-$ by $\texttt{node}^{\tau,+}$ and $\texttt{node}^{\tau,-}$, respectively, and analogously for the link- and graph-level features.

While the node- and link-level features are similar to the heuristics of counting walks, $\texttt{graph}^{\tau,+}$ and $\texttt{graph}^{\tau,-}$ are not directly useful for link prediction as discussed in the introduction: the information related to the presence of $\{i, j\}$ is obfuscated by the summation over the entire subgraph (by taking the trace).

SEAL attempts to remedy a similar issue for average (as opposed to $\text{tr}(\mathbf{P}^\tau)$) pooling by labeling nodes by distance from the link. Here we propose a principled alternative. Since transition probabilities are normalized and have clear topological meaning, we can suppress irrelevant information by measuring the perturbation of graph-level features, $\Delta \texttt{graph}^\tau = \texttt{graph}^{\tau,+} - \texttt{graph}^{\tau,-}$. This "background subtraction" induces a data-driven soft limit on the longest loop length so that, by design, $\Delta \texttt{graph}^\tau$ concentrates around the focal link. In Appendix E, we demonstrate locality of WALKPOOL. Compared to node labeling, the perturbation approach does not manually assign a relative position (which may wrongly suggest that nodes at the same distance from the candidate link are of equal importance).

In summary, with WALKPOOL, for all $\mathcal{G} \in \{\mathcal{G}_{\{i,j\}} : \{i, j\} \in \mathcal{E}^c\}$, we read a list of features as

$$
\texttt{WP}_\theta(G, \mathbf{Z}) = \left[ \omega_{1,2}, \left( \texttt{node}^{\tau,+}, \texttt{node}^{\tau,-}, \texttt{link}^{\tau,+}, \texttt{link}^{\tau,-}, \Delta \texttt{graph}^\tau \right)_{\tau=2}^{\tau_c} \right], \quad (4)
$$

where $\tau_c$ is the cutoff of the walk length and we treat it as a hyperparameter. The features are finally fed into a classifier; we use an MLP $\Pi_\theta$ with a sigmoid activation in the output. The ablation study in Table 5 (Appendix B) shows that all the computed features contribute relevant predictive information. Walk profile computation is summarized in Figure 2.

### 3.4 Training the model

The described model contains trainable parameters $\theta$ which are fitted on the given observed set $\mathcal{E}^t$ of positive and negative training links and their enclosing subgraphs. We train WALKPOOL with MSE loss (see Appendix G for a discussion of the loss),

$$
\theta^* = \arg\min_\theta \frac{1}{|\mathcal{E}^t|} \sum_{\{i,j\} \in \mathcal{E}^t} \left( y_{\{i,j\}} - \Pi_\theta \left( \texttt{WP}_\theta(\mathcal{G}_{\{i,j\}}, \mathbf{Z}_{\{i,j\}}) \right) \right)^2
$$

where $y_{\{i,j\}} = 1$ if $\{i, j\} \in \mathcal{E}^o$ and 0 otherwise is the label indicating whether the link $\{i, j\}$ is true or false. The fitted model is then deployed on the candidate links $\mathcal{E}^c$; the predicted label for $\{x, y\} \in \mathcal{E}^c$ is simply $\widehat{y}_{\{x,y\}} = \Pi_{\theta^*}(\texttt{WP}_{\theta^*}(\mathcal{G}_{\{x,y\}}), \mathbf{Z}_{\{x,y\}})$.

## 4 Performance of WalkPool on benchmark datasets

We use area under the curve (AUC) (Bradley, 1997) and average precision (AP) as metrics. Precision is the fraction of true positives among predictions. Letting TP (FP) be the number of true (false) positive links, $\text{AP} = \text{TP}/(\text{TP} + \text{FP})$.

### 4.1 Datasets

In homophilic (heterophilic) graphs, nodes that are similar (dissimilar) are more likely to connect. For node classification, a homophily index is defined formally as the average fraction of neighbors with identical labels (Pei et al., 2020). In the context of link prediction, if we assume that network formation is driven by the similarity (or dissimilarity) of node attributes, then a homophilic graph will favor triangles while a heterophilic graph will inhibit

triangles. Following this intuition, we adopt the average clustering coefficient `ACC` (Watts & Strogatz, 1998) as a topological correlate of homophily.

We experiment with eight datasets without node attributes and seven with attributes. As graphs without attributes we use: (i) **USAir** (Batagelj & Mrvar, 2006), (ii) **NS** (Newman, 2006), (iii) **PB** (Ackland et al., 2005), (iv) **Yeast** (Von Mering et al., 2002), (v) **C.ele** (Watts & Strogatz, 1998), (vi) **Power** (Watts & Strogatz, 1998), (vii) **Router** (Spring et al., 2002), and (viii) **E.coli** (Zhang et al., 2018). Properties and statistics of the datasets, including number of nodes and edges, edge density and `ACC` can be found in Table 4 of Appendix A.

In graphs with a very low average clustering coefficient like **Power** and **Router** (`ACC` = 0.080 and 0.012, respectively, see Appendix A), topology-based heuristics usually perform poorly (Lü & Zhou, 2011) (cf. Table 2), since heuristics often adopt the homophily assumption and rely on triangles. We show that by *learning* the topological organizing patterns, WALKPOOL performs well even for these graphs.

For a fair comparison, we use the exact same training and testing sets (including positive and negative links) as SEAL in Zhang & Chen (2018). 90% of edges are taken as positive training edges and the remaining 10% are the positive test edges. The same number of additionally sampled nonexistent links are taken as training and testing negative edges.

As graphs *with* node attributes, we use: (i) **Cora** (McCallum et al., 2000), (ii) **Citeseer** (Giles et al., 1998), (iii) **Pubmed** (Namata et al., 2012), (iv) **Chameleon** (Rozemberczki et al., 2021), (v) **Cornell** (Craven et al., 1998), (vi) **Texas** (Craven et al., 1998), and (vii) **Wisconsin** (Craven et al., 1998). The properties and statistics of the datasets can be found in Table 4 of Appendix A; further details are provided in Appendix A

Following the experimental protocols in (Kipf & Welling, 2016b; Pan et al., 2018; Mavromatis & Karypis, 2020), we split the links in three parts: 10% testing, 5% validation, 85% training. We sample the same number of nonexisting links in each group as negative links.

## 4.2 BASELINES

On benchmarks without node attributes, we compare WALKPOOL with eight other methods. We consider three walk-based heuristics: AA, Katz and PR; two subgraph-based heuristic learning methods: Weisfeiler–Lehman graph kernel (WLK) (Shervashidze et al., 2011) and WLNM (Zhang & Chen, 2017); and latent feature methods: node2vec (N2V) (Grover & Leskovec, 2016), spectral clustering (SPC) (Tang & Liu, 2011), matrix factorization (MF) (Koren et al., 2009) and LINE (Tang et al., 2015). We additionally consider the GNN-based SEAL, which is the previous state-of-the-art.

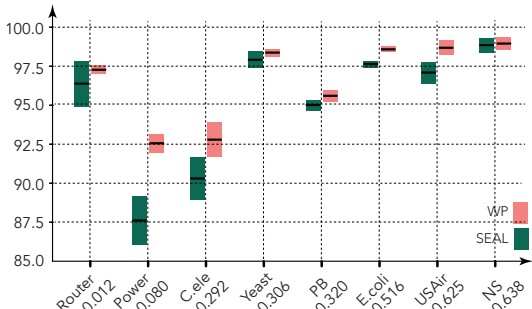

Figure 4: Comparison of mean and variance of AUC between SEAL and WP with 90% observed links. The datasets are sorted by their clustering coefficients.

For datasets with node attributes, we combine WALKPOOL with three unsupervised GNN-based models: the VGAE, adversarially regularized variational graph autoencoder (ARGVA) (Pan et al., 2018) and Graph Info-Clust (GIC) (Mavromatis & Karypis, 2020).

## 4.3 IMPLEMENTATION DETAILS

In the absence of node attributes, we initialize the node representation $\mathbf{Z}^{(0)}$ as an $N \times F^{(0)}$ all-ones matrix, with $F^{(0)} = 32$. We generate node representations by a two-layered Graph Convolutional Network (GCN) (Kipf & Welling, 2016a): $\mathbf{Z}^{(k)} = \text{GCN}\left(\mathbf{Z}^{(k-1)}\right), k \in \{1, 2\}$, where $\mathbf{Z}^{(0)} = \mathbf{X}$. We then concatenate the outputs as $\mathbf{Z} = [\mathbf{Z}^{(0)} \mid \mathbf{Z}^{(1)} \mid \mathbf{Z}^{(2)}]$ to obtain the final node representation used to compute $\text{WP}(G, \mathbf{Z})$ and classify. While a node labeling scheme like the one in SEAL is not needed for the strong performance of WALKPOOL, we

| | WP | SEAL | AA | CN | VGAE |
|---|---|---|---|---|---|
| Triangle lattice ($50 \times 50$) | 99.77±0.12 | 99.54±0.11 | 95.16±0.90 | 95.15±0.87 | 37.14±1.47 |
| 2D-grid ($50 \times 50$) | 99.86±0.09 | 99.51±0.09 | 49.90±0.09 | 49.90±0.09 | 32.43±1.12 |
| Hypercube ($2^{10}$) | 100.00±0.00 | 99.60±0.10 | 48.01±0.35 | 48.01±0.35 | 37.84±1.19 |
| Star (1000) | 100.00±0.00 | 99.57±0.10 | 14.50±2.40 | 14.50±2.40 | 100.00±0.00 |

Table 1: AUC for synthetic graphs over 10 independent trials.

find that labeling nodes by distance to focal link slightly improves results; we thus evaluate WALKPOOL with and without distance labels.

For the three datasets with node attributes, we first adopt an unsupervised model to generate an initial node representation $\mathbf{Z}^{(0)}$. This is because standard 2-layer GCNs are not expressive enough to extract useful node features in the presence of node attributes Zhang et al. (2019).

The initial node representation is fed into the same two-layered GCN above, and we take the concatenated representation $\mathbf{Z} = [\mathbf{Z}^{(0)} \mid \mathbf{Z}^{(1)} \mid \mathbf{Z}^{(2)}]$ as the input to WALKPOOL. The concatenation records local multi-hop information and facilitates training by providing skip connections. It gives a small improvement over only using the last-layer embeddings, i.e., $\mathbf{Z} = \mathbf{Z}^{(2)}$. The hyperparameters of the model are explained in detail in Appendix C. The runtime of WALKPOOL can be found in Appendix H.

### 4.4 RESULTS

**Synthetic datasets**  To show that WALKPOOL successfully learns topology, we use four synthetic datasets: **Triangle lattice** (ACC = 0.415), **2D-Grid** (ACC = 0), **Hypercube** (ACC = 0), and **Star** (ACC = 0). We randomly remove 10% links from each graph and run link prediction algorithms. AUC for WALKPOOL, SEAL, AA, and CN is shown in Table 1. In these regular graphs the topological organizing rules are known explicitly. It is therefore clear that a common-neighbor-based heuristic (such as CN or AA) should fail for **2D-grid** since none of the connected nodes have common neighbors. WalkPool successfully learns the organizing patterns without prior knowledge about the graph formation rules, using the same hyperparameters as in all other experiments. For **Triangle lattice** and **2D-grid**, WalkPool achieves a near-100% AUC. Small errors are due to hidden the test edges which act as noise; for fewer withheld links the error would vanish. In **hypercube** and **Star**, WalkPool achieves an AUC of 100% in all ten trials. The **star** graph is heterophilic in the sense that no triangles are present; we indeed observe that AA and CN have AUC below 50% since they (on average) flip true and false edges.

**Datasets without attributes**  We perform 10 random splits of the data. The average AUCs with standard deviations are shown in Table 2; the AP results and statistical significance of the results can be found in Appendix D. For WALKPOOL, we have considered both the cases with and without the DL node labels as input features.

From Table 2, WALKPOOL significantly improves the prediction accuracy on all the datasets we have considered. It also has a smaller standard deviation, indicating that WALKPOOL is stable on independent data splits (unlike competing learning methods). Among all methods, WALKPOOL with DL performs the best; the second-best method slightly below is WALKPOOL without DL. In other words, although DRNL slightly improves WALKPOOL, nonetheless, WALKPOOL achieves SOTA performance already without it.

WALKPOOL performs stably on both homophilic (high ACC) and heterophilic (low ACC) datasets, achieving state-of-the-art performance on all. Remarkably, on **Power** and **Router** where topology-based heuristics such as AA and Katz fail, WALKPOOL shows strong performance (it also outperforms SEAL by about 5% on **Power**). This confirms that walk profiles are expressive descriptors of the local network organizing patterns, and that WALKPOOL can fit their parameters from data without making prior topological assumptions. Experiments with 50% observed training edges can be found in Appendix F.

**Datasets with node attributes**  We apply WALKPOOL to extract higher-order information from node representations generated via unsupervised learning. We again perform 10 random splits of the data and report the average AUC with standard deviations in Table 3; for AP see Appendix D. In Table 3, we show the results of unsupervised models with and

| Data | USAir | NS | PB | Yeast | C.ele | Power | Router | E.coli |
|------|-------|-----|-----|-------|-------|-------|--------|--------|
| AA | 95.06±1.03 | 94.45±0.93 | 92.36±0.34 | 89.43±0.62 | 86.95±1.40 | 58.79±0.88 | 56.43±0.51 | 95.36±0.34 |
| Katz | 92.88±1.42 | 94.85±1.10 | 92.92±0.35 | 92.24±0.61 | 86.34±1.89 | 65.39±1.59 | 38.62±1.35 | 93.50±0.44 |
| PR | 94.67±1.08 | 94.89±1.08 | 93.54±0.41 | 92.76±0.55 | 90.32±1.49 | 66.00±1.59 | 38.76±1.39 | 95.57±0.44 |
| WLK | 96.63±0.73 | 98.57±0.51 | 93.83±0.59 | 95.86±0.54 | 89.72±1.67 | 82.41±3.43 | 87.42±2.08 | 96.94±0.29 |
| WLNM | 95.95±1.10 | 98.61±0.49 | 93.49±0.47 | 95.62±0.52 | 86.18±1.72 | 84.76±0.98 | 94.41±0.88 | 97.21±0.27 |
| N2V | 91.44±1.78 | 91.52±1.28 | 85.79±0.78 | 93.67±0.46 | 84.11±1.27 | 76.22±0.92 | 65.46±0.86 | 90.82±1.49 |
| SPC | 74.22±3.11 | 89.94±2.39 | 83.96±0.86 | 93.25±0.40 | 51.90±2.57 | 91.78±0.61 | 68.79±2.42 | 94.92±0.32 |
| MF | 94.08±0.80 | 74.55±4.34 | 94.30±0.53 | 90.28±0.69 | 85.90±1.74 | 50.63±1.10 | 78.03±1.63 | 93.76±0.56 |
| LINE | 81.47±10.71 | 80.63±1.90 | 76.95±2.76 | 87.45±3.33 | 69.21±3.14 | 55.63±1.47 | 67.15±2.10 | 82.38±2.19 |
| SEAL | 97.09±0.70 | 98.85±0.47 | 95.01±0.34 | 97.91±0.52 | 90.30±1.35 | 87.61±1.57 | 96.38±1.45 | 97.64±0.22 |
| WP(ones) | 98.52±0.50 | 98.86±0.42 | 95.42±0.39 | 98.16±0.33 | 92.42±1.22 | 91.71±0.60 | 97.18±0.28 | 98.54±0.20 |
| WP(DL) | **98.68±0.48** | **98.95±0.41** | **95.60±0.37** | **98.37±0.25** | **92.79±1.09** | **92.56±0.60** | **97.27±0.28** | **98.58±0.19** |

Table 2: Prediction accuracy measured by AUC on eight datasets (90% observed links) without node attributes. Boldface marks the best, underline the second best results.

| | VGAE | | ARGVA | | GIC | |
|---|------|----|-------|----|-----|----|
| | NO WP | WP | NO WP | WP | NO WP | WP |
| **Cora** | 91.98±0.54 | 94.64±0.55 | 92.45±1.11 | 94.71±0.85 | 93.68±0.59 | **95.90±0.50** |
| **Citeseer** | 91.21±1.14 | 94.32±0.90 | 91.71±1.38 | 94.53±1.77 | 95.03±0.65 | **95.94±0.53** |
| **Pubmed** | 96.51±0.14 | 98.49±0.13 | 96.62±0.12 | 98.52±0.14 | 93.00±0.36 | **98.72±0.10** |
| **Chameleon** | 98.79±0.18 | 99.51±0.08 | 98.23±0.24 | 99.51±0.08 | 94.13±0.38 | **99.52±0.08** |
| **Cornell** | 70.59±9.03 | 78.24±7.51 | 81.73±4.82 | **82.39±8.92** | 63.32±7.47 | 80.69±7.25 |
| **Texas** | 73.71±9.29 | **76.02±7.05** | 68.05±8.29 | 75.62±5.80 | 65.43±10.39 | 74.49±6.85 |
| **Wisconsin** | 75.05±6.88 | 77.07±6.11 | 75.69±7.91 | 79.34±6.32 | 74.74±6.28 | **82.27±6.27** |

Table 3: Prediction accuracy (AUC) on datasets with node attributes (90% observed links).

without WALKPOOL. For all the unsupervised models and datasets, WALKPOOL improves the prediction accuracy. On the **Pubmed** dataset where the relative importance of topology over features is greater, WALKPOOL brings about the most significant gains.

While traditional heuristics cannot include node attributes, WALKPOOL encodes the node attributes into random walk transition probabilities via the attention mechanism, which allows it to capture structure and node attributes simultaneously. The prediction accuracy relies on the unsupervised GNN which generates the inital node representation. We find that the combination of GIC and WALKPOOL yields the highest accuracy for all three datasets. Importance of initial node representations is plain since WALKPOOL is not designed to un-cover structure in node attributes. Nonetheless, WALKPOOL greatly improves performance of unsupervised GNN on the downstream link prediction task.

## 5 DISCUSSION

The topology of a graph plays a much more important role in link prediction than node classification, even with node attributes. Link prediction and topology are entangled, so to speak, since topology is *defined* by the links. Most GNN-based link prediction methods work with node representations and do not adequately leverage topological information.

Our proposed WALKPOOL, to the contrary, jointly encodes node representations and graph topology into *learned* topological features. The central idea is how to leverage learning: we apply attention to the node representations and interpret the attention coefficients as transition probabilities of a graph random walk. WALKPOOL is a trainable topological heuristic, thus explicitly considering long-range correlations, but without making ad hoc assumptions like the classical heuristics. This intuition is borne out in practice: combining supervised or unsupervised graph neural networks with WALKPOOL yields state-of-the-art performance on a broad range of benchmarks with diverse structural characteristics. Remarkably, WALKPOOL achieves this with the same set of hyperparameters on all tasks regardless of the network type and generating mechanism.

### ACKNOWLEDGMENTS

LP would like to acknowledge support from National Natural Science Foundation of China under Grand No. 62006122. CS and ID were supported by the European Research Council (ERC) Starting Grant 852821—SWING.

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

## Appendix A. Benchmark datasets description

Brief description of the benchmark datasets is as follows. For the datasets without node attributes, we have considered:

- **USAir** (Batagelj & Mrvar, 2006): a graph of US airlines with 332 nodes and 2126 edges; $\texttt{ACC} = 0.625$.
- **NS** (Newman, 2006): the collaboration relation of network science researchers with 1589 nodes and 2742 edges; $\texttt{ACC} = 0.638$.
- **PB** (Ackland et al., 2005): a graph of hyperlinks between weblogs on US politics with 1222 nodes and 16714 edges; $\texttt{ACC} = 0.320$.
- **Yeast** (Von Mering et al., 2002): a protein-protein interaction graph in yeast with 2375 nodes and 11693 edges; $\texttt{ACC} = 0.306$.
- **C.ele** (Watts & Strogatz, 1998): the biological neural network of C.elegans with 297 nodes and 2148 edges; $\texttt{ACC} = 0.292$.
- **Power** (Watts & Strogatz, 1998): the topology of the Western States Power Grid of the United States with 4941 nodes and 6594 edges; $\texttt{ACC} = 0.080$.
- **Router** (Spring et al., 2002): the router-level Internet with 5022 nodes and 6258 edges; $\texttt{ACC} = 0.012$.
- **E.coli** (Zhang et al., 2018): the pairwise reaction relation of metabolites in E.coli with 1805 nodes and 15660 edges; $\texttt{ACC} = 0.516$.

As graphs *with* node attributes, we consider three citation graphs with publications described by binary vectors indicating the absence/presence of the corresponding dictionary word:

- **Cora** (McCallum et al., 2000): a citation graph with 2708 scientific publications and 5278 links. The dictionary consists of 1433 unique words; $\texttt{ACC} = 0.241$.
- **Citeseer** (Giles et al., 1998): the dataset consists of 3312 scientific publications with 4552 links; the dictionary consists of 3703 unique words; $\texttt{ACC} = 0.141$.
- **Pubmed** (Namata et al., 2012): the dataset consists of 19717 scientific publications with 44324 links. The dictionary consists of 500 unique words; $\texttt{ACC} = 0.060$.

We also consider a Wikipedia page graph where nodes represent web pages and edges represent hyperlinks between them. Node features represent several informative nouns in the pages:

- **Chameleon** (Rozemberczki et al., 2021): Wikipedia page-page graph under the topic chameleon. The graph consists of 2277 nodes and 31371 edges where each node has an attribute vector of dimension 2325; $\texttt{ACC} = 0.377$.

Finally, we consider three webpage graphs which include web pages from computer science departments of various universities, node features are the bag-of-words representation of web pages.

- **Cornell** (Craven et al., 1998): a webpage graph with 183 nodes and 277 edges, and the node attribute has dimension 1703; $\texttt{ACC} = 0.167$.
- **Texas** (Craven et al., 1998): a webpage graph with 183 nodes and 279 edges, and the node attribute has dimension 1703; $\texttt{ACC} = 0.198$.
- **Wisconsin** (Craven et al., 1998): a webpage graph with 251 nodes and 450 edges, and the node attribute has dimension 1703; $\texttt{ACC} = 0.208$.

The benchmark dataset properties and statistics are summarized in Table 4.

## Appendix B. Ablation study

We conduct ablation studies of WalkPool by excluding or including only each of the node-, link- and graph-level features in (4). We used short notations $\{\texttt{node}^\tau\} \equiv \{\texttt{node}^{\tau,+}, \texttt{node}^{\tau,-} :$

| Dataset | USAir | NS | PB | Yeast | C.ele | Power | Router | E.coli | Cora | Citeseer | Pubmed | Chameleon | Cornell | Texas | Wisconsin |
|---------|-------|-----|-----|-------|-------|-------|--------|--------|------|----------|--------|-----------|---------|-------|-----------|
| Node | 332 | 1589 | 1222 | 2375 | 297 | 4941 | 5022 | 1805 | 2708 | 3312 | 19717 | 2277 | 183 | 183 | 251 |
| Edges | 2126 | 2742 | 16714 | 11693 | 2148 | 6594 | 6258 | 15660 | 5278 | 4552 | 44324 | 31371 | 227 | 279 | 450 |
| ACC | 0.625 | 0.638 | 0.320 | 0.306 | 0.292 | 0.080 | 0.012 | 0.516 | 0.241 | 0.141 | 0.060 | 0.377 | 0.167 | 0.198 | 0.208 |
| Density | 3.86e-2 | 2.17e-3 | 2.24e-2 | 4.15e-3 | 4.87e-2 | 5.40e-4 | 4.96e-4 | 9.61e-3 | 1.44e-3 | 8.30e-4 | 2.28e-4 | 1.21e-2 | 1.35e-2 | 1.67e-2 | 1.42e-2 |
| Attributed | No | No | No | No | No | No | No | No | Yes | Yes | Yes | Yes | Yes | Yes | Yes |

Table 4: Benchmark dataset properties and statistics.

$\tau \in \{2, \cdots, \tau_c\}\}$, and similarly for $\{\texttt{link}^\tau\}$, $\{\Delta\texttt{graph}^\tau\}$. The ablation results for the C.ele dataset is shown in the In Table. 5.

| Feature `feat` | $\omega_{12}$ | $\{\texttt{node}^\tau\}$ | $\{\texttt{link}^\tau\}$ | $\{\Delta\texttt{graph}^\tau\}$ | - |
|----------------|---------------|--------------------------|--------------------------|---------------------------------|---|
| AUC (using only `feat`) | 87.90 | 91.88 | 92.44 | 92.29 | - |
| AUC (using WP\`feat`) | 92.69 | 92.66 | 91.08 | 92.65 | 92.82 |

Table 5: Ablation study in C.ele

## APPENDIX C. HYPERPARAMTERS

The hyperparamters to reproduce the results are summarized in Table 6.

For the split of training, validation and testing edges, we have adopted different setups for datasets with and without node attributes to consist with previous studies. In particular, for datasets without node attributes, 90% of edges are taken as training positive edges and the remaining 10% are for testing positive edges. The corresponding same number of negative edges are sampled randomly for training and testing. Then among the training edges. we randomly selected 5% for validation. The observed graph from which the enclosing subgraphs are sampled consists of all the training positive edges. For the datasets with node attributes, all the edges are divided into 85% for training, 10% for testing and 5% for validation. The observed graph is built only based on the training edges. Some studies of link prediction choose to keep the observed graph connected when sampling the test edges, while others do not adopt this option. For the experiment results shown in the paper, we sample the test edges uniformly random without ensuring the observed graph is connected.

For sampling the enclosing subgraphs, we set the number of hops as 2 except 3 for the Power dataset. The 2-hop subgraphs sampled from E.coli and PB datasets and Pubmed are relatively numerous in nodes. Thus we set a maximum number nodes per hop for these two datasets to fit into memory. In particular, for each hop during sampling, we randomly select a maximum of 100 nodes if the sampled nodes exceeds.

For WALKPOOL, We set $\tau_c = 7$ for the cutoff of walk length in all experiments. When $\tau_c$ becomes large, the transition probabilities converges to a constant, as the random walk reaches stationary. Therefore, a larger value of $\tau$ introduces little further information. From experiments, we find that introducing a larger $\tau_c$ will not increase the accuracy notably.

For the classifer, we use a MLP with layer sizes $72 - 1440 - 1440 - 720 - 72 - 1$. We use Relu as the activation function for the hidden layers, and sigmoid function for the output layer.

## APPENDIX D. RESULTS MEASURED BY AP AND STATISTICAL SIGNIFICANCE OF RESULTS

The prediction accuracy measured by AP for datasets with and without node attributes are shown in Table 8 and Table 9, respectively. From the tables, WALKPOOL still performs best with the AP metric.

To verified the statistical significance of the results, we performed a two-sided hypothesis test with the null hypothesis that two independent samples (corresponding to the results of WalkPool and the second best algorithm) have identical average. We compute the corresponding $p$-value on a per dataset basis for the eight datasets without node attributes. The results are shown in Table 7. The second best algorithm is SEAL except in C.ele (where

| Name | With attributes | No attributes |
|---|---|---|
| optimizer | Adam | Adam |
| learning rate | 5e-5 | 5e-5 |
| weight decay | 0 | 0 |
| test ratio | 0.1 | 0.1 |
| validation ratio | 0.05 of training edges | 0.05 of all edges |
| batch size | 32 | 32 |
| epochs | 50 | 50 |
| hops of enclosing subgraph | $(*)$ 2 | 2 |
| dimension of initial representation $\mathbf{Z}^{(0)}$ | 16 | 32 |
| initial representation $\mathbf{Z}^{(0)}$ | ones or DL | unsupervised models |
| hidden layers of GCN | 32 | 32 |
| output layers of GCN | 32 | 32 |
| hidden layers of attention MLP | 32 | 32 |
| output layers of attention MLP | 32 | 32 |
| walk length cutoff $\tau_c$ | 7 | 7 |
| heads | 2 | 2 |

Table 6: Default hyperparameters for reproducing the reults. $(*)$: 3-hop for the Power dataset.

| Data | USAir | NS | PB | Yeast | C.ele | Power | Router | E.coli |
|---|---|---|---|---|---|---|---|---|
| Second Best | 97.09±0.70 | 98.85±0.47 | 95.01±0.34 | 97.91±0.52 | 90.32±1.49 | 91.78±0.61 | 96.38±1.45 | 97.64±0.22 |
| WP(DL) | 98.68±0.48 | 98.95±0.41 | 95.60±0.37 | 98.37±0.25 | 92.79±1.09 | 92.56±0.60 | 97.27±0.28 | 98.58±0.19 |
| $p$-value | $2.18 \cdot 10^{-5}$ | $6.18 \cdot 10^{-1}$ | $1.60 \cdot 10^{-3}$ | $2.56 \cdot 10^{-2}$ | $6.00 \cdot 10^{-4}$ | $9.90 \cdot 10^{-3}$ | $8.68 \cdot 10^{-2}$ | $7.80 \cdot 10^{-9}$ |

Table 7: $p$-value by comparing WP and second best algorithm on eight datasets with no attributes.

| Data | USAir | NS | PB | Yeast | C.ele | Power | Router | E.coli |
|---|---|---|---|---|---|---|---|---|
| AA | 95.36±1.00 | 94.46±0.93 | 92.36±0.46 | 89.53±0.63 | 86.46±1.43 | 58.76±0.89 | 56.50±0.51 | 96.05±0.25 |
| Katz | 94.07±1.18 | 95.05±1.08 | 93.07±0.46 | 95.23±0.39 | 85.93±1.69 | 79.82±0.91 | 64.52±0.81 | 94.83±0.30 |
| PR | 95.08±1.16 | 95.11±1.04 | 92.97±0.77 | 95.47±0.43 | 89.56±1.57 | 80.56±0.91 | 64.91±0.85 | 96.41±0.33 |
| WLK | 96.82±0.84 | 98.79±0.40 | 93.34±0.89 | 96.82±0.35 | 88.96±2.06 | 83.02±3.19 | 86.59±2.23 | 97.25±0.42 |
| WLNM | 95.95±1.13 | 98.81±0.49 | 92.69±0.64 | 96.40±0.38 | 85.08±2.05 | 87.16±0.77 | 93.53±1.09 | 97.50±0.23 |
| N2V | 89.71±2.97 | 94.28±0.91 | 84.79±1.03 | 94.90±0.38 | 83.12±1.90 | 81.49±0.86 | 68.66±1.49 | 90.87±1.48 |
| SPC | 78.07±2.92 | 90.83±2.16 | 86.57±0.61 | 94.63±0.56 | 62.07±2.40 | 91.00±0.58 | 73.53±1.47 | 96.08±0.37 |
| MF | 94.36±0.79 | 78.41±3.85 | 93.56±0.71 | 92.01±0.47 | 83.63±2.09 | 53.50±1.22 | 82.59±1.38 | 95.59±0.31 |
| LINE | 79.70±11.76 | 85.17±1.65 | 78.82±2.71 | 90.55±2.39 | 67.51±2.72 | 56.66±1.43 | 71.92±1.53 | 86.45±1.82 |
| SEAL | 97.13±0.80 | 99.06±0.37 | 94.55±0.43 | 98.33±0.37 | 89.48±1.85 | 89.55±1.29 | 96.23±1.71 | 98.03±0.20 |
| WP(ones) | 98.43±0.66 | 99.04±0.28 | 95.00±0.46 | 98.52±0.28 | 91.14±1.80 | 92.45±0.72 | 97.08±0.42 | 98.74±0.17 |
| WP(DL) | **98.66±0.55** | **99.09±0.29** | **95.28±0.41** | **98.64±0.28** | **91.53±1.33** | **93.07±0.69** | 97.20±0.38 | **98.79±0.21** |

Table 8: Prediction accuracy measured by AP on eight datasets (90% observed links) without node attributes. Boldface letters are used to mark the best results while underlined letters indicate the second best results.

the second best is PR) and Power (where the second-best is SPC). Recall that a $p$-value of 0.05 or less is customarily considered statistically significant. We see that for all but the NS and Router datasets the $p$-value is below 0.05. For most datasets it is orders of magnitude below. The AUC on the NS dataset is already very close to 100, thus leaving little space for improvement; for Router it is the large variance of SEAL that gives a $p$-value a bit above 0.05. Note that even for Router and NS the empirical mean of WalkPool is better. More trials should easily break the statistical tie even in those cases, but we used the same 10 splits as SEAL for a fair comparisons.

|  | VGAE | | ARGVA | | GIC | |
|---|---|---|---|---|---|---|
|  | NO WP | WP | NO WP | WP | NO WP | WP |
| **Cora** | 92.65±0.59 | 95.11±0.53 | 93.11±1.08 | 95.23±0.84 | 93.45±0.48 | **95.97±0.57** |
| **Citeseer** | 92.28±0.92 | 94.89±0.89 | 92.69±0.85 | 95.04±1.46 | 95.11±0.65 | **96.04±0.63** |
| **Pubmed** | 96.60±0.13 | 98.46±0.14 | 96.52±0.20 | 98.49±0.14 | 92.32±0.37 | **98.65±0.15** |
| **Chameleon** | 98.79±0.22 | **99.50±0.15** | 98.19±0.28 | 99.48±0.15 | 93.34±0.50 | 99.46±0.15 |
| **Cornell** | 75.34±8.24 | 82.74±7.89 | 84.66±5.48 | **85.92±7.45** | 66.94±8.20 | 84.18±8.39 |
| **Texas** | 78.50±8.18 | **81.78±5.75** | 73.10±8.26 | 81.21±5.00 | 70.89±9.08 | 79.60±6.34 |
| **Wisconsin** | 79.56±6.32 | 82.50±4.96 | 78.72±6.24 | 82.25±8.10 | 78.27±4.78 | **84.45±4.60** |

Table 9: Prediction accuracy measured by AP on seven datasets (90% observed links) with node attributes.

APPENDIX E. LOCALITY OF GRAPH-LEVEL FEATURES

Let $G = (\mathcal{V}, \mathcal{E})$ be an enclosing subgraph, and $\{1, 2\}$ be the candidate link. We denote the random walk transition matrices on $G^+$ and $G^-$ as $\mathbf{P}$ and $\mathbf{Q}$, respectively. For any node $x, y \in \mathcal{V}$, let $d(x, y)$ be the shortest path distance on the graph. We define the distance from $x$ to the candidate link to be $\bar{d}(x) = \min\{d(x, 1), d(x, 2)\}$. We have the following result.

**Proposition 1.** *Let* $\mathcal{V}^\tau = \{x \in \mathcal{V}, \bar{d}(x) > \tau\}$, *then* $[\mathbf{P}^\tau]_{x,y} = [\mathbf{Q}^\tau]_{x,y}$ *for all* $x, y \in \mathcal{V}^\tau$.

*Proof.* We prove the result via induction. When $\tau = 1$, after the node-wise normalization from the softmax function in (2), the elements of $\mathbf{P}$ and $\mathbf{Q}$ are identical among nodes that not neighbors of the candidate link, i.e., $\mathcal{V}^1 = \{x \in \mathcal{V}, \bar{d}(x) > 1\}$.

Now suppose the claim holds for $\tau$. Let $\mathcal{I} = \{x \in \mathcal{V}, \bar{d}(x) \leq \tau\}$, let $\partial\mathcal{I}$ be the set of nodes that are adjacency to but not in $\mathcal{I}$, and let $\mathcal{I}^c$ be the set of rest nodes in the graph. We can arrange the order of the nodes that such that $\mathbf{P}$ and $\mathbf{P}^\tau$ are of the following block form

$$\mathbf{P} = \begin{pmatrix} \mathbf{P}_{\mathcal{I},\mathcal{I}} & \mathbf{P}_{\mathcal{I},\partial\mathcal{I}} & \mathbf{0} \\ \mathbf{P}_{\partial\mathcal{I},\mathcal{I}} & \mathbf{P}_{\partial\mathcal{I},\partial\mathcal{I}} & \mathbf{P}_{\partial\mathcal{I},\mathcal{I}^c} \\ \mathbf{0} & \mathbf{P}_{\mathcal{I}^c,\partial\mathcal{I}} & \mathbf{P}_{\mathcal{I}^c,\mathcal{I}^c} \end{pmatrix}, \qquad \mathbf{P}^\tau = \begin{pmatrix} \mathbf{P}^\tau_{\mathcal{I},\mathcal{I}} & \mathbf{P}^\tau_{\mathcal{I},\partial\mathcal{I}} & \mathbf{P}^\tau_{\mathcal{I},\mathcal{I}^c} \\ \mathbf{P}^\tau_{\partial\mathcal{I},\mathcal{I}} & \mathbf{P}^\tau_{\partial\mathcal{I},\partial\mathcal{I}} & \mathbf{P}^\tau_{\partial\mathcal{I},\mathcal{I}^c} \\ \mathbf{P}^\tau_{\mathcal{I}^c,\mathcal{I}} & \mathbf{P}^\tau_{\mathcal{I}^c,\partial\mathcal{I}} & \mathbf{P}^\tau_{\mathcal{I}^c,\mathcal{I}^c,} \end{pmatrix} \qquad (5)$$

where $\mathbf{P}_{\mathcal{I},\mathcal{I}}$ is the block matrix confined to $\mathcal{I}$ and other blocks are defined similarly. By definition, $\mathbf{P}_{\mathcal{I},\mathcal{I}^c} = \mathbf{0}$ and $\mathbf{P}_{\mathcal{I}^c,\mathcal{I}} = \mathbf{0}$. Multiplication of the block matrices gives

$$\mathbf{P}^{\tau+1}_{\mathcal{I}^c,\mathcal{I}^c} = \mathbf{P}_{\mathcal{I}^c,\partial\mathcal{I}}\mathbf{P}^\tau_{\partial\mathcal{I},\mathcal{I}^c} + \mathbf{P}_{\mathcal{I}^c,\mathcal{I}^c}\mathbf{P}^\tau_{\mathcal{I}^c,\mathcal{I}^c}. \qquad (6)$$

We can do a similar computation to obtain $\mathbf{Q}^{\tau+1}_{\mathcal{I}^c,\mathcal{I}^c}$, with all $\mathbf{P}$ replaced by $\mathbb{Q}$ in the above equation. By assumption, we have

$$\mathbf{P}_{\mathcal{I}^c\partial\mathcal{I}} = \mathbf{Q}_{\mathcal{I}^c\partial\mathcal{I}} \qquad \mathbf{P}^\tau_{\partial\mathcal{I},\mathcal{I}^c} = \mathbf{Q}^\tau_{\partial\mathcal{I},\mathcal{I}^c} \qquad \mathbf{P}_{\mathcal{I}^c,\mathcal{I}^c} = \mathbf{Q}_{\mathcal{I}^c,\mathcal{I}^c}, \qquad \mathbf{P}^\tau_{\mathcal{I}^c,\mathcal{I}^c} = \mathbf{Q}^\tau_{\mathcal{I}^c,\mathcal{I}^c}, \qquad (7)$$

as $\partial\mathcal{I}, \mathcal{I}^c \in \mathcal{V}^\tau$, therefore we obtain $\mathbf{P}^{\tau+1}_{\mathcal{I}^c,\mathcal{I}^c} = \mathbf{Q}^{\tau+1}_{\mathcal{I}^c,\mathcal{I}^c}$. As $\mathcal{I}^c = \mathcal{V}^{\tau+1}$, the claim follows by induction. $\square$

APPENDIX F. RESULTS WITH 50% OBSERVATION

We further test of the performance of WALKPOOL under the setup of sparse training set. In particular, we keep only 50% positive links of the graphs for training and use the rest links for testing. The same number of negative links are sampled randomly for the traning set and test set. The results measure by AUC and AP are shown in Table 10 and 11, respectively.

APPENDIX G. MSE LOSS AND BCE LOSS

As link prediction is a classification problem, we usually adopt a classification loss such as binary cross-entropy (BCE). We opted for MSE based on the following heuristic. In node classification, categories are usually clearly defined. For example, in a citation graph, the category of a paper is near-definite. Meanwhile, the topology of real graphs is often fuzzy and evolving. In a co-authorship graph, some authors who have not published together today

| Data | USAir | NS | PB | Yeast | C.ele | Power | Router | E.coli |
|------|-------|-----|-----|-------|-------|-------|--------|--------|
| AA | 88.61±0.40 | 77.13±0.75 | 87.06±0.17 | 82.63±0.27 | 73.37±0.80 | 53.38±0.22 | 52.94±0.28 | 87.66±0.56 |
| Katz | 88.91±0.51 | 82.30±0.93 | 91.25±0.22 | 88.87±0.28 | 79.99±0.59 | 57.34±0.51 | 54.39±0.38 | 89.81±0.46 |
| PR | 90.57±0.62 | 82.32±0.94 | 92.23±0.21 | 89.35±0.29 | 84.95±0.58 | 57.34±0.52 | 54.44±0.38 | 92.96±0.43 |
| WLK | 91.93±0.71 | 87.27±1.71 | 92.54±0.33 | 91.15±0.35 | 83.29±0.89 | 63.44±1.29 | 71.25±4.37 | 92.38±0.46 |
| WLNM | 91.42±0.95 | 87.61±1.63 | 90.93±0.23 | 92.22±0.32 | 75.72±1.33 | 64.09±0.76 | 86.10±0.52 | 92.81±0.30 |
| N2V | 84.63±1.58 | 80.29±1.20 | 79.29±0.67 | 90.18±0.17 | 75.53±1.23 | 55.40±0.84 | 62.45±0.81 | 84.73±0.81 |
| SPC | 65.42±3.41 | 79.63±1.34 | 78.06±1.00 | 89.73±0.28 | 47.30±0.91 | 56.51±0.94 | 53.87±1.33 | 92.00±0.50 |
| MF | 91.28±0.71 | 62.95±1.03 | 93.27±0.16 | 84.99±0.49 | 78.49±1.73 | 50.53±0.60 | 77.49±0.64 | 91.75±0.33 |
| LINE | 72.51±12.19 | 65.96±1.60 | 75.53±1.78 | 79.44±7.90 | 59.46±7.08 | 53.44±1.83 | 62.43±3.10 | 74.50±11.10 |
| SEAL | 93.36±0.67 | 90.88±1.18 | 93.79±0.25 | 93.90±0.54 | 82.33±2.31 | 65.84±1.10 | 86.64±1.58 | 94.18±0.41 |
| WP(ones) | 95.16±0.70 | 90.68±1.04 | 94.50±0.20 | 94.89±0.22 | **87.83±0.83** | 67.03±0.77 | 88.09±0.52 | **95.37±0.22** |
| WP(DL) | **95.50±0.74** | **90.97±0.96** | **94.57±0.16** | **95.00±0.21** | 87.62±1.39 | **67.72±0.86** | 88.13±0.61 | 95.33±0.30 |

Table 10: Prediction accuracy measured by AUC on eight datasets (50% observed links) without node attributes. Boldface letters are used to mark the best results while underlined letters indicate the second best results.

| Data | USAir | NS | PB | Yeast | C.ele | Power | Router | E.coli |
|------|-------|-----|-----|-------|-------|-------|--------|--------|
| AA | 89.39±0.39 | 77.14±0.74 | 87.24±0.18 | 82.68±0.27 | 73.40±0.77 | 53.37±0.23 | 52.94±0.27 | 89.01±0.49 |
| Katz | 91.29±0.36 | 82.69±0.88 | 91.54±0.16 | 92.22±0.21 | 79.94±0.79 | 57.63±0.52 | 60.87±0.26 | 91.93±0.35 |
| PR | 91.93±0.50 | 82.73±0.90 | 91.92±0.25 | 92.54±0.23 | 84.15±0.86 | 57.61±0.56 | 61.01±0.30 | 94.68±0.28 |
| WLK | 93.34±0.51 | 89.97±1.02 | 92.34±0.34 | 93.55±0.46 | 83.20±0.90 | 63.97±1.81 | 75.49±3.43 | 94.51±0.32 |
| WLNM | 92.54±0.81 | 90.10±1.11 | 91.01±0.20 | 93.93±0.20 | 76.12±1.08 | 66.43±0.85 | 86.12±0.68 | 94.47±0.21 |
| N2V | 82.51±2.08 | 86.01±0.87 | 77.21±0.97 | 92.45±0.23 | 72.91±1.74 | 60.83±0.68 | 66.77±0.57 | 85.41±0.94 |
| SPC | 70.18±2.16 | 81.16±1.26 | 81.30±0.84 | 92.07±0.27 | 55.31±0.93 | 59.10±1.06 | 59.13±3.22 | 94.14±0.29 |
| MF | 92.33±0.90 | 66.62±0.89 | 92.53±0.33 | 87.28±0.57 | 77.82±1.59 | 52.45±0.63 | 81.25±0.56 | 94.04±0.36 |
| LINE | 71.75±11.85 | 71.53±0.97 | 78.72±1.24 | 83.06±9.70 | 60.71±6.26 | 55.11±3.49 | 64.87±6.76 | 75.98±14.45 |
| SEAL | 94.15±0.54 | 92.21±0.97 | 93.42±0.19 | 95.32±0.38 | 81.99±2.18 | 65.28±1.25 | 87.79±1.71 | 95.67±0.24 |
| WP(ones) | 95.39±0.73 | 92.15±0.81 | 94.14±0.27 | 96.04±0.16 | **86.49±0.97** | 69.26±0.64 | **89.21±0.44** | 96.35±0.24 |
| WP(DL) | **95.87±0.74** | **92.33±0.76** | **94.22±0.27** | **96.15±0.13** | 86.25±1.42 | **69.79±0.71** | 89.17±0.55 | **96.36±0.34** |

Table 11: Prediction accuracy measured by AP on eight datasets (50% observed links) without node attributes. Boldface letters are used to mark the best results while underlined letters indicate the second best results.

may have submitted a paper that will come out tomorrow. We therefore expect a less peaky distribution of link probabilities than node classes, and we choose a loss that minimizes the miscalibration of the model. In this context where we want to predict probabilities as accurately as possible, the MSE loss is known as the Brier score (another textbook use of the Brier score is to calibrate the chance-of-rain forecasts).

We have experimented with both the binary cross-entropy (BCE) loss and the MSE loss, and we observed no discernible difference. For example, for the eight datasets without node attributes (the last line of Table 2), accuracies measured by AUC when using MSE/BCE loss are: 98.68/98.68; 98.95/98.85; 95.60/95.69; 98.37/98.37; 92.79/92.83; 92.56/92.58; 97.27/97.35; 98.58/98.67.

## Appendix H. Runtime of WalkPool

The runtime of WalkPool is similar to that of SEAL when using the same number of hops to construct the subgraphs. On a classical dataset **USAir**, WalkPool takes 129.62s for 50 epochs with 1-hop subgraph sampling, while SEAL takes 145.94s. The configuration of WalkPool used throughout our paper has two GCN layers followed by several linear layers for computing the powers of matrix $\mathbf{P}$, and a four-layer MLP classifier. As such, from the perspective of runtime, the trainable part of WalkPool architecture is no more complex than that of a typical GNN. For very large datasets, the most time-consuming step is to extract the enclosing subgraphs—this is true for WalkPool and any other subgraph-based link prediction algorithm.

