# OpenReview forum: "Neural Link Prediction with Walk Pooling"
_ICLR.cc/2022/Conference — ICLR 2022 Poster_

### Official Review · Reviewer_MHUH · 2021-11-02

**Correctness:** 3
**Technical Novelty And Significance:** 3
**Empirical Novelty And Significance:** 3
**Recommendation:** 8
**Confidence:** 4

**Main Review:**

Strengths:
- The paper provides an elegant formulation of the link prediction problem. The execution of the proposed methodology is also interesting.
- The paper is clearly written, presenting the proposed model in a rigorous way but at the same time providing good motivation.
- The experimental evaluation is quite thorough.

Weaknesses:
-	Lack experiments on benchmark (artificial) graphs. This could further increase our understanding of the functioning of the model
-	The baseline models could be enhanced. Although the list is quite complete, some of the used models are not tailored for the link prediction task.

**Detailed comments.**

As I mentioned above, I really enjoyed reading the paper. Despite building upon ideas presented in  SEAL, the formulation is interesting. Besides, the arguments made seem to be valid. Yet, some points are not very clear and need further elaboration. Next, I summarize key points related to the paper.

To begin with, although the paper uses standard benchmarks for link prediction, I would expect to have some experiments on artificial datasets in order to better understand the functioning of the model. As I can see, most of the networks used have really high clustering coefficient---which is not the case for the majority of real-world graphs. Having artificial networks where one could control ACC would be very helpful.

In the case of the absence of node attributes, the paper concatenates the outputs of the three-layer to obtain the final node representation. Aren’t the embeddings of the final layer enough for this? Please elaborate more on this aspect.

Although spectral clustering has been widely used before, I think it is not an appropriate baseline for link prediction. I would rather be in favor of using other unsupervised feature learning models (e.g., LINE, NetMF).

The datasets used are quite small in terms of the number of nodes and edges, and there is no discussion about the running time of the proposed model. How does WalkPool’s running time compare against SEAL?



**Summary Of The Paper:**

The paper introduces WalkPool, a model for link prediction that relies on GNNs. The main idea is to jointly encode node representations and graph topology information into node features which could be learned in an end-to-end-manner. The proposed problem formulation is interesting: the transition probabilities to capture proximity among nodes are computed on top of a latent graph which is the outcome of an attention mechanism on node features. The proposed methodology is presented clearly. Besides, the authors have performed experiments on various graphs, and the performance of WalkPool is compared against state-of-the-art baseline models.

**Summary Of The Review:**

Overall, I believe it’s an interesting approach that offers a new perspective on the link prediction task. Despite building upon ideas presented in previous work (SEAL), the paper makes progress in presenting an elegant methodology. Some aspects related to the experimental evaluation should be addressed.

---

> ### Author Response · Authors · 2021-11-18
> **Spectral clustering works well on "Power"**
>
> **[Multipart response (part 2/2)]**
>
> *Although spectral clustering has been widely used before, I think it is not an appropriate baseline for link prediction. I would rather be in favor of using other unsupervised feature learning models (e.g., LINE, NetMF).*
>
> **Response**
>
> We agree that spectral clustering is not specially designed for link prediction; it usually performs worse than other unsupervised feature learning models. The reason we included it in the results is that in the dataset **Power** (with a **very low** ACC $0.08$), the spectral method works better than any other algorithm but WalkPool. WalkPool automatically learns this low-clustering structure (topology) and thus performs well. Per your suggestion we have included the results of Line and NetMF in the revision.
>
> *The datasets used are quite small in terms of the number of nodes and edges, and there is no discussion about the running time of the proposed model. How does WalkPool’s running time compare against SEAL?*
>
> **Response**
>
> The runtime of WalkPool is similar to that of SEAL when using the same number of hops to construct the subgraphs. On a classical dataset **USAir**, WalkPool takes 129.62s for 50 epochs with 1-hop subgraph sampling, while SEAL takes 145.94s. The configuration of WalkPool used throughout our paper has two GCN layers followed by several linear layers for computing the powers of matrix $\mathbf{P}$, and a four-layer MLP classifier. As such, from the perspective of runtime, the trainable part of WalkPool architecture is no more complex than that of a typical GNN. For very large datasets, the most time-consuming step is to extract the enclosing subgraphs---this is true for both SEAL and WalkPool. We added these remarks about runtimes and complexities to Appendix H.
>
>
> Summarizing, we hope that the new experiments and related discussions adequately address your comment that _Some aspects related to the experimental evaluation should be addressed_.

---

> > ### Comment · Reviewer_MHUH · 2021-12-06
> > **Thank you for the revision**
> >
> > I would like to thank the authors for addressing my comments. Enhancing the experimental evaluation with synthetic graphs and additional baselines was important to further demonstrate the performance of the proposed methodology. Overall I believe the paper makes a good contribution to the problem of link prediction with GNNs. I will keep my score as is, voting for acceptance.

---

> ### Author Response · Authors · 2021-11-18
> **Thank you for the strong endorsement + new synthetic graphs**
>
> **[Multipart response (part 1/2)]**
>
> Thank you for your positive feedback and concrete suggestions. We are very pleased that you enjoyed the paper.
>
> *To begin with, although the paper uses standard benchmarks for link prediction, I would expect to have some experiments on artificial datasets in order to better understand the functioning of the model. As I can see, most of the networks used have really high clustering coefficient---which is not the case for the majority of real-world graphs.*
>
> **Response:**
>
> To better illustrate the idea, in the revised manuscript we describe experiments on four artificial graphs: **triangle lattice** ($\mathtt{ACC}=0.415$), **2D-grid** ($\mathtt{ACC}=0$), **hypercube** ($\mathtt{ACC}=0$), and **star** ($\mathtt{ACC}=0$). The results can be found at the beginning of Section 4.4. We randomly remove 10\% links from each graph and run link prediction algorithms. Prediction accuracy measured by AUC is shown in Table 1.
>
> In these regular graphs the topological organizing rules are known explicitly. It is therefore clear that a common-neighbor-based heuristic (such as CN or AA) should fail for **2D-grid** since none of the connected nodes have common neighbors. WalkPool successfully learns the topology patterns of these graphs without any prior knowledge about the graph formation rules, using the same hyperparameters as in all other experiments. For **triangular lattice** and **2D-grid**, WalkPool achieved a near-100\% AUC. Small errors are due to the hidden test edges which act as noise; with fewer withheld links the error would vanish. In **hypercube** and **star**, WalkPool achieves an AUC of 100\% in _all ten trials_. The **star** graph is heterophilic in the sense that no triangles are present; we indeed observe that AA and CN have AUC below 50\% since they (on average) flip true and false edges.
>
> *Having artificial networks where one could control ACC would be very helpful.*
>
> **Response:**
>
> Following your suggestion we looked into graph models with continuously controllable ACC such as the Watts-Strogatz (WS) small-world model. The WS model starts from a $k$-nearest neighbor graph and gradually randomizes the edges. Alas, this randomization makes the graph inherently ``unpredictable,'' preventing us from properly testing link prediction algorithms. Other models like the _exponential graph model_ have the same problem: the graph becomes unpredictable when ACC is low.
>
> We hope that the new lattice examples which span the qualitative extremes of homophily and heterophily adquately illustrate that our model performs well also for low ACC.
> If the referee is aware of ACC-tunable graph families that are more suitable for link prediction studies we would welcome the suggestion and certainly include it in the next update.
>
> *In the case of the absence of node attributes, the paper concatenates the outputs of the three-layer to obtain the final node representation. Aren’t the embeddings of the final layer enough for this? Please elaborate more on this aspect.*
>
> **Response:**
>
> The concatenation plays two roles: it records multi-hop local information around nodes  and it facilitates training by providing a kind of skip connections. The same approach has been adopted in GNN architeture designs such as DGCNN (Zhang et al. "An end-to-end deep learning architecture for graph classification." Thirty-Second AAAI Conference on Artificial Intelligence. 2018.) We experimented with using only the last-layer embeddings. Keeping other hyperparameters the same, the AUCs with/without the concatenation are: 98.68±0.48/98.50±0.53 in USAir,  92.56±0.60/92.46±0.59 in Power, 92.79±1.09/92.68±1.09 in C.ele, and 98.95±0.41/98.89±0.38 in NS. Indeed, while there are clear improvements with the concatenation, they are not extraordinary. We added a remark to this effect in the last paragraph of Section 4.3.

---

> > ### Author Response · Authors · 2021-11-30
> > **thank you for suggesting synthetic graphs!**
> >
> > Dear reviewer MHUH,
> >
> > We'd like to thank you one more time for a detailed and helpful review.
> >
> > In particular, your suggestion to experiment with synthetic datasets resulted in a new batch of experiments that we are very happy about. They serve as a great backdrop for the introductory discussions but also demonstrate that WalkPool indeed works well on a very wide range of graphs (achieving 100% AUC on some datasets)
> >
> > The discussion window is almost closed but in this very last minute we want to state that if there are any other concerns remaining we would be happy to address them.
> >
> > It would be great to hear your thoughts about all the updates and discussions.
> >
> > Sincerely,
> >
> > The authors

---

### Official Review · Reviewer_ye2u · 2021-11-03

**Correctness:** 3
**Technical Novelty And Significance:** 3
**Empirical Novelty And Significance:** 2
**Recommendation:** 6
**Confidence:** 4

**Main Review:**

Strengths:
1. Building the latent subgraph from embeddings is an interesting idea.
2. The extract features somehow encodes path information which may be useful for link prediction.
3. Comprehensive experiments are conducted.

Weaknesses:
1. Motivations in the Introduction section are not clearly stated.

**Summary Of The Paper:**

The paper proposes a new link prediction method called WP, based on treating link prediction as a subgraph classification problem, where the link of interest is surrounded by the subgraph. Existing GNN architectures could be used for learning node embeddings. With node embeddings, a latent subgraph could be built using self-attention, where attention scores are used for computing the subgraph edge weights. With the latent subgraph, a set of features could be computed to be used as the features for the link of interest. The features will be used for subsequent link prediction. Essentially, the proposed WP method is a feature extractor for links (or its surrounding subgraph).

**Summary Of The Review:**

The paper proposes a pooling method (i.e., a latent structure feature extractor) for learning features that describe links (i.e., the surrounding subgraphs of the links). Building the latent subgraph from embeddings turns continuous information back to discrete graph space, which facilitates feature extraction. The framework remains end-to-end learnable. The extracted features summarizes multi-hop path information, which somehow reflect higher order structural information. Experiments on multiple real datasets are conducted.

In general, the paper proposes a neat idea with good experiment design. The current draft could be further improved by revising its Introduction section, as many claims are not clear and do not seem to motivate the method design. For example, Figure 1 does not provide much information about the idea. Also, I think the word "topology" does not well describe the proposed idea.

---

> ### Author Response · Authors · 2021-11-18
> **Thank you for the positive assessment + motivation in Introduction section + explain Figure 1 + use of "topology"**
>
>
> Thank you for the positive assessment. We tried our best to address your comments in the updated manuscript. We hope but are not 100 percent sure that we correctly interpreted your points. If not, we would welcome your further input.
>
> *Motivations in the Introduction section are not clearly stated. For example, Figure 1 does not provide much information about the idea.*
>
> **Response:**
>
> Prompted by your remark we expanded the motivating discussions in the introduction. We added an explicit motivation statement. We also emphasized the difference between our method and standard GNNs in the last paragraph of Section 3.2 to describe the difference between the proposed approach and previous GNN models.
> We also agree that Figure 1 could be better motivated. Our intention was to show two topologically distinct graphs---one very homophilic in the sense that it prefers triangles, or common neighbors, and one very heterophilic in the sense that there are absolutely no common neighbors, on which many traditional heuristics with hard-wired topological preference will fail.  Prompted by yours and other reviews, we updated the paper with a completely new batch of experiments on artificial graphs which includes the two extremes shown in Figure 1. We have accordingly expanded the explanations related to the Figure in the introduction and we added a pointer to Section 4.4 where the new experiments with lattices are given.
>
> *Also, I think the word "topology" does not well describe the proposed idea.*
>
> **Response:**
>
> You are right that in mathematics, _topology_ refers to properties of shapes that are preserved under continuous deformations, which are global properties. Our use of "topology" for (local) connectivity patterns or local structure is different, but it seems to be standard in the network literature (for example Newman, M. (2018). Networks. Oxford University Press or Albert, Réka, and Albert-László Barabási. "Statistical mechanics of complex networks." Reviews of Modern Physics 74.1 (2002): 47.). In WalkPool, transition probabilities are not purely combinatorial counts but rather "topologically interpretable counts" which jointly encode graph structure and node attributes; this is different from GNNs where they are embedded into node representations. Thus we use "topology" to mean "topologically-interpretable features / fingerprints". We have clarified this point in a footnote in page 1 of the revised manuscript.
>
> If you feel strongly that a better term exists we would be happy to consider it. Also: we may be missing your point in which case we'd welcome further input.

---

> > ### Author Response · Authors · 2021-11-30
> > **New discussions and new experiments**
> >
> > Dear reviewer ye2u,
> >
> > Thank you again for taking the time to review our manuscript. In the process of addressing yours and the comments of other reviewers we added a number of new experiments and clarifications. In particular, related to your remark that Figure 1 and the corresponding description in the introduction are unclear, we added a new batch of experiments on synthetic regular grids that we are very excited about. We believe they illustrate the principles behind WalkPool and provide an intuitive backdrop for Figure 1 and the related descriptions. They also corroborate our claims about state-of-the-art performance.
> >
> > It would be great to hear what you think, especially if you have other concerns that we did not address.
> >
> > Sincerely,
> >
> > The authors

---

### Official Review · Reviewer_C9p9 · 2021-11-03

**Correctness:** 3
**Technical Novelty And Significance:** 3
**Empirical Novelty And Significance:** 3
**Recommendation:** 6
**Confidence:** 4

**Main Review:**

Strengths:

The paper is well-written; the introduction of the previous work is organized. Description of the SEAL algorithm is very insightful and clear.

Weaknesses:

The attributed network datasets concsidered is a bit limited. Can there be more datasets available for evaluation?

The final features extracted are mainly specified by human. Though effective, it leaves an open question that whether thses sorts of features can be leared adaptively?

The proposed method follows the framework of the SEAL algorithm, and the authors should exphasize the differences between the two methods and what makes the propose method more advantageous.

**Summary Of The Paper:**

This paper proposes a neural architecture for predicting missing links in a graph by designing a so-called WalkPool scheme. WalkPool exploits node features (from a GNN) to compute the transition probability between the nodes, and then manually design a number of links that are extracted from the oder-k encolosing sub-graphs covering a pair of nodes for link prediction. The authors also defined a delta-graph feature that is the sub-graph level feature difference before and after removing the node pairs with which the link is to be predicted. Experimental results are quite optimistic on about 10 benchmark datasets.

**Summary Of The Review:**


Overall the paper is nicely written and organized, and provides useful thought in the problem of link prediction with or without node features. Experimental results are also promising. However I have a few technical details I would like to discuss.

(1) The authors used feature-based similarty (like in GAT type of algorithms) to define the transition probbaility matrix, which takes the place of the topology-based transitiona matrix by and large. Is there a way to combine the information from both?

(2) The pooling part is a bit confusing to me, in that it seems that the WALKPOOL indeed does not perform pooling but instead use pre-defined features, while pooling is usually for a set (of objects such as nodes) with varying numbe of elements. Can authors add more details of the pooling used in the SEAL and why WALKPOOL is considered a pooling scheme?

---

> ### Author Response · Authors · 2021-11-18
> **Thank you for your positive assessment + new attributed graphs + adaptive feature learning + difference between SEAL +  combine $\mathbf{A}$ and $\mathbf{P}$**
>
>
> Thank you for the positive assessment and clear pointers to possible improvements.
>
> *The attributed network datasets concsidered is a bit limited. Can there be more datasets available for evaluation?*
>
> **Response:**
>
> We tested on commonly used datasets with attributes for link prediction, but we agree with the reviewer that there is an imbalance with those without attributes. In the updated manuscript we added results for four other datasets with attributes, namely **Chameleon**, **Cornell**, **Texas** and **Wisconsin**. The results can be found in Table~3. WalkPool exhibits strong state-of-the-art performance on the new datasets, too.
>
> *The final features extracted are mainly specified by human.  Though effective, it leaves an open question whether these sorts of features can be  learned adaptively?*
>
>
> **Response:**
>
> That is an interesting question. Topological features are inherently discrete. Learning the right topological features might entail differentiable programming with some continuous surrogates. We believe that this may further improve our strong results but it would entail  additional research. One option would be to look at learnable linear functionals on $\mathbf{P}^\tau$; we ran preliminary experiments with this idea but they didn't pan out, possibly due to overfitting. Nonetheless, the idea is intriguing and we will keep thinking about it. As a partial justification, many learning algorithms are a combination of some fixed discrete choices (e.g., the number of layers, the dimensions of feature maps, the number and size of filters in a CNN) with some trainable parts.
>
>
> *The proposed method follows the framework of the SEAL algorithm, and the authors should exphasize the differences between the two methods and what makes the propose method more advantageous*
>
> **Response:**
>
> We indeed adopt the subgraph classification approach to link prediction as proposed in SEAL (as we state on the first paragraph of Section 1.2 in the manuscript). We note in the passing that subgraphs for GNN-based link prediction can be found in earlier works (e.g.  (Zhang  \&  Chen,  2017)), which we cite in paragraph 3 of Section 1.1. Using subgraphs has a long history in link prediction, as many localized heuristics can be interpreted as using subgraphs implicitly (Lü \& Zhou, 2011). This has been opportunely leveraged by SEAL.
>
> To the best of our understanding, one key contribution of SEAL is the distance-related  node labeling  (DRNL) for link prediction. This is further emphasized by the authors of SEAL in a follow-up paper (Zhang et al. "Revisiting graph neural networks for link prediction." arXiv:2010.16103 (2020).) The main difference is in how the two algorithms encode and leverage graph topology. SEAL adds DRNL labeling to node attributes, and then applies a GNN to generate a node embedding which is used for classification after the so-called sort pooling. We explicitly calculate salient topological features such as probabilities of triangular or quadrangular walks around the focal link. Although DRNL slightly improves our method, WalkPool achieves SOTA performance already without it, as shown in the last two columns of Table 2 in the updated manuscript. We added a clarification of this point in Section 4, and futher emphasized it in the introduction under our contributions.
>
> *The authors used feature-based similarty (like in GAT type of algorithms) to definethe  transition  probbaility  matrix,  which  takes  the  place  of  the  topology-basedtransitiona matrix by and large.  Is there a way to combine the information fromboth?*
>
> **Response:**
> Indeed, there is, and in an early version we worked with combinations of $\mathbf{A}$ and $\mathbf{P}$. This gave small improvements in accuracy in the best case, but it introduced additional hyperparameters without clear nominal values.
>
> *The pooling part is a bit confusing to me, in that it seems that the WALKPOOL indeed does not perform pooling but instead use pre-defined features, while pooling is usually for a set (of objects such as nodes) with varying numbe of elements. Can authors add more details of the pooling used in the SEAL and why WALKPOOL is considered a pooling scheme?*
>
> **Response:**
> Our use of the word pooling is perhaps a bit liberal. As the reviewer points out, pooling often refers to some form of coarsening (e.g., in CNNs) the output of which is an object of the same type (e.g, a downsampled image). The last layer of a classification CNN or GNN involves global average pooling. Average pooling can be seen as a graph summarization mechanism. Our idea is to replace average pooling by a different graph summarization mechanism which retains topological information. Hence we use the term pooling since WalkPool is meant to replace the standard average pooling of features. We added a clarification footnote about our use of ``pooling'' on page 5. We hope that you find this argument reasonable.

---

> > ### Author Response · Authors · 2021-11-30
> > **Are the new graphs with attributes useful?**
> >
> > Dear reviewer C9p9,
> >
> > Thank you again for helping us improve the manuscript. We were wondering whether the new experiments we ran prompted by your suggestion address your concern about a small number of graphs with attributes? There are now seven such graphs in total (with WalkPool yielding the best performance on all). Further, if you still still have concerns about the terminology, we could welcome your input. We cannot change the paper now but we can try and address them in comments and commit to the requisite changes later.
> >
> > Any further input would certainly be valuable.

---

> ### Author Response · Authors · 2021-12-09
> **At the risk of being annoying, may we ask whether our responses and additional experiments on datasets with attributes speak to your initial concerns and suggestions?**
>
> Dear Reviewer C9p9,
>
> At the risk of being annoying, may we ask whether our responses and additional experiments on datasets with attributes speak to your initial concerns and suggestions?
>
> Best wishes,
>
> The authors

---

### Official Review · Reviewer_yAGw · 2021-11-08

**Correctness:** 2
**Technical Novelty And Significance:** 1
**Empirical Novelty And Significance:** 1
**Recommendation:** 5
**Confidence:** 4

**Main Review:**

Strengths:
(1) The paper is clearly written
(2) I find the walk based approach to be interesting (with several weaknesses)

Weakness:
No conceptual difference with GNN: In my understanding, when we compute $P^{\tau}$, it combines the $\tau$ hop information around a node to compute its embedding. In this aspect, I did not understand any conceptual difference with GNN. What aspect of it is making the difference?

Loss function: At the end, LP is a classification task. From this aspect, MSE loss minimization may not be suitable--- a simple classification or ranking loss should be more suitable. Did the authors try them?

AUC: I suspect that for most of the datasets, the difference of WALKPOOL with baselines is not statistically significant. Can the authors clarify about it?



**Summary Of The Paper:**

The paper provides a method for link prediction based on random walk based pooling method (WalkPool). It aggregates the probabilities of the adjacent structure of a node or link to design the node embeddings. Finally, it shows superiority of the method against baselines via AUC and AP.

**Summary Of The Review:**

The walk based approach is interesting but the paper lacks novelty, clear articulation about high or low level difference of WalkPool with GNN. Also, MSE  loss is chosen for a clear classification task. The results are weak too.

---

> ### Author Response · Authors · 2021-11-18
> **Statistical significance of WalkPool**
>
> **[Multipart response (part 2/2)]**
>
> *I suspect that for most of the datasets, the difference of WALKPOOL with baselines is not statistically significant. Can the authors clarify about it?*
>
> **Response:**
>
> We were at first confused (since we achieve state of the art results for all datasets, on most by a large margin), but we realized that the issue might have arisen from the way we structured Table 2, where the second-best algorithm is a simpler version of WalkPool which indeed performs very closely to the full WalkPool, and not the previous SOTA. The actual second-best previous algorithm is (mostly) SEAL, and the difference between SEAL and WalkPool is significant. The reason to include the simplified WalkPool is to show that the bulk of the improvement brought about by WalkPool is _not_ due to DRNL. To hopefully reduce this confusion and aid visualization, we separated out the two versions of WalkPool in Tables 2, 8, 9 and 10 by a horizontal line.
>
> Further, motivated by your comment, we performed a two-sided hypothesis test with the null hypothesis that two independent samples (corresponding to the results of WalkPool and the second best algorithm) have identical average. We compute the corresponding $p$-value on a per dataset basis for the eight datasets without node attributes. The results are as follows:
>
> |**Data**|**USAir**|**NS**|**PB**|**Yeast**|**C.ele**|**Power**|**Router**|**E.coli**|
> |----------|----------|----------|----------|----------|----------|----------|----------|----------|
> |SecondBest|$97.09±0.70$|$98.85±0.47$|$95.01±0.34$|$97.91±0.52$|$90.32±1.49$|$91.78±0.61$|$96.38±1.45$|$97.64±0.22$|
> |WP(DL)|$98.68±0.48$|$98.95±0.41$|$95.60±0.37$|$98.37±0.25$|$92.79±1.09$|$92.56±0.60$|$97.27±0.28$|$98.58±0.19$|
> |$p$-value|$2.18\cdot10^{-5}$|$6.18\cdot10^{-1}$|$1.60\cdot10^{-3}$|$2.56\cdot10^{-2}$|$6.00\cdot10^{-4}$|$9.90\cdot10^{-3}$|$8.68\cdot10^{-2}$|$7.80\cdot10^{-9}$|
>
>
> and also added in Appendix D in the updated manuscript. The second best algorithm is SEAL except in C.ele (where the second best is PR) and Power (where the second-best is SPC).
> Recall that a $p$-value of $0.05$ or less is customarily considered statistically significant. We see that for all but the NS and Router datasets the $p$-value is below $0.05$. For most datasets it is **orders of magnitude** below. The AUC on the NS dataset is already very close to $100$, thus leaving little space for improvement; for Router it is the large variance of SEAL that gives a $p$-value a bit above $0.05$. Note that even for Router and NS the empirical mean of WalkPool is better. We are confident that more trials would easily break the statistical tie even in those cases, but we used the same 10 splits as SEAL for a fair comparisons.
>
> Further, improvements brought about by WalkPool are more significant than those brought about by SEAL over previous state of the art. For example, on the E.coli dataset, SEAL improved the previous SOTA (WLNM) from $97.21$ to $97.64$, while the AUC of WalkPool is $98.58$. For the USAir dataset, SEAL improved previous SOTA (WLK) from $96.63$ to $97.09$, while WalkPool achieves AUC $98.68$.
>
> Finally, beyond improvements on a per-dataset basis, the strength of WalkPool is that it performs well **on all datasets**.
>
> We can thus safely claim that WalkPool improves over existing methods in a **highly statistically significant manner**, more so than those methods improved over previous state of the art.
>
> To conclude, with regard to your summary that "The walk based approach is interesting but the paper lacks novelty, clear articulation about high or low level difference of WalkPool with GNN. Also, MSE loss is chosen for a clear classification task. The results are weak too.", we hope that our responses clearly show that WalkPool is rather different from GNNs, that MSE is not an unreasonable choice, and that we achieve state-of-the-art results on a broad range of benchmarks, with improvements that are statistically significant (new Table 7).

---

> > ### Author Response · Authors · 2021-11-30
> > **Thoughts about our responses**
> >
> > Dear reviewer yAGw,
> >
> > As the discussion period is about to close any minute now, we were wondering whether you had a chance to evaluate our responses to your comments? One of your concerns was about the statistical significance of improvements. We feel that the tiny $p$-values and the fact that we achieve state of the art performance on all datasets are strong arguments in favor of statistical significance (dare we say high statistical significance). We elaborate on this in detail in the response above.
> >
> > We also have experiments that show that for any imaginable combination of losses between baselines and WalkPool, WalkPool always achieves the state of the art performance. And if your concerns about the differences between GNNs and WalkPool persist after our response, we may be still able to expand on unclear aspects.
> >
> > In any case, it would be great to hear your thoughts.
> >
> > Sincerely,
> >
> > The authors

---

> ### Author Response · Authors · 2021-11-18
> **Thank you for the feedback + difference with GNN + loss function**
>
> **[Multipart response (part 1/2)]**
>
>
> We thank the reviewer for their valuable feedback. We respectfully disagree with most of the critical points raised (and we hope to articulate why in the following), but we feel that they helped us improve the manuscript.
>
>
> *No conceptual difference with GNN: In my understanding, when we compute $P^\tau$, it combines the  hop information around a node to compute its embedding. In this aspect, I did not understand any conceptual difference with GNN. What aspect of it is making the difference?*
>
> **Response:**
>
> GNNs and WalkPool perform different tasks:
>
> - GNNs use powers of the adjacency matrix $\mathbf{A}^\tau$ to compute graph filters (or generalized weighted combinations of neighborhood attributes in GAT and message passing nets). The input to a GNN is the graph topology (encoded by $\mathbf{A}$) and node features; the output is a processed set of node features or node embeddings (a function on nodes).
> - WalkPool uses the output of a GNN (or any other node embedding algorithm) to effectively define **a new weighted graph** (Equation (1)); the output of this step are effective *edge weights*. It then uses those weights to define a random walk on the graph; the output of this step are edge traversal probabilities (Equation (2)). This is very different from a GNN. Even at this point, WalkPool does not use those probabilities to compute an update of node features like a GNN (it would be odd to do it with a row-normalized $\mathbf{P}$) but it operates directly on the entries of $\mathbf{P}^\tau$ to explicitly extract probabilities of topology-revealing motifs (such as loops of various lengths).
> - This is the key difference between WalkPool and the previous link prediction algorithms, all of which directly use node embeddings. SEAL does better by adding distance to the focal link to node features. This improves results but it still only indirectly (if at all) encodes features such as triangle frequencies which are known to play a key role in link predictability.
> - We note that almost all graph algorithms (beyond GNNs) are specified in terms of some graph matrices (adjacency $\mathbf{A}$, random walk $\mathbf{P}$, Laplacian $\mathbf{L}$) and their powers. This is the most economical language to describe graph algorithms and it does not invalidate their merit.
>
> Prompted by your comment, to make these important points clearer in the revised manuscript, we added the following at the end of Section 3.2: "The use of  $\mathbf{P}$ in WalkPool is different from how graph matrices (e.g., $\mathbf{A}$) are used in GNNs. In GNNs, the powers $\mathbf{A}^{\tau}$ serve as shift operators between neighborhoods that are multiplied by filter weights and used to weigh node features; WalkPool encodes graph signals into effective edge weights and directly extracts topological information from the entries of $\mathbf{P}^\tau$."
>
> *(Loss function) At the end, LP is a classification task. From this aspect, MSE loss minimization may not be suitable---a simple classification or ranking loss should be more suitable. Did the authors try them?*
>
> **Response:**
>
> We have indeed experimented with both the binary cross-entropy (BCE) loss and the MSE loss and we observed no discernible difference. For example, for the eight datasets without node attributes (the last line of Table 2 in the updated manuscript), accuracies when using MSE/BCE loss are:  98.68/98.68;  98.95/98.85; 95.60/95.69; 98.37/98.37; 92.79/92.83; 92.56/92.58; 97.27/97.35; 98.58/98.67.
>
> We opted for MSE based on the following heuristic. In node classification categories are usually clearly defined. For example, in a citation graph, the category of a paper is near-definite. Meanwhile, the topology of real graphs is often fuzzy and evolving. In a co-authorship graph, some authors who have not published together today may have submitted a paper that will come out tomorrow. We therefore expect a less peaky distribution of link probabilities than node classes, and we choose a loss that minimizes the miscalibration of the model. In this context where we want to predict _probabilities_ as accurately as possible, the MSE loss is known as the Brier score (another use of the Brier score is to calibrate the chance-of-rain forecasts). That said, again, we did not observe a real difference between the two losses. To clarify this we added the above results for the classification loss in Appendix G and an explanatory footnote to the main text.

---

### Author Response · Authors · 2021-11-18
**Summary of revisions in round one**

Dear reviewers,

Thank you for the useful feedback and insightful questions. We uploaded a revision of the manuscript that implements many of your suggestions and we are adding the detailed point-by-point responses below. The main changes and additions are:

- Experiments on four new datasets with node attributes
- Addition of new synthetic benchmarks with known organizing rules (on all new datasets WalkPool outperforms prior art)
- Computation of $p$-values to establish statistical significance of improvements
- Expanded motivations in the introduction and pointers to new synthetic benchmarks
- Many added explanations and justifications of terminology

All changes and additions are typeset in purple. We are looking forward to your further feedback.

Sincerely,

The authors

---

### Author Response · Authors · 2021-11-29
**Do the new experiments and discussions address your concerns?**

Dear reviewers,

Thank you again for all your work. In response to your comments we ran many new experiments and thought hard about how to improve our explanations. We feel that this resulted in a better paper. We are particularly happy about the new experiments on synthetic datasets where WalkPool sometimes achieves a 100% AUC. We find these experiments to be an illuminating illustration of WalkPool and we are grateful to you for suggesting them.

Do these new experiments and explanations address your concerns? There is only a little time left to discuss but if your concerns remain unaddressed we may still try to expand our responses. It would be great to hear your thoughts.

Sincerely,
The authors

---

### Decision · Program_Chairs · 2022-01-20

**Decision:**

Accept (Poster)

**Comment:**

This paper proposes a new link prediction algorithm based on a pooling scheme called WalkPool. The main idea is to jointly encode node representations and graph topology information into node features and conduct the learning end-to-end. The paper shows the superiority of the method against the baselines.

Strength
* The paper is generally clearly written.
* A new method is proposed, which is technically sound.
* Many experiments are conducted to verify the effectiveness of the proposed method.

Weakness
* The novelty of the work might not be so significant.  There is a similarity with the SEAL algorithm.

The authors have addressed most of the problems pointed out by the reviewers. They have also conducted additional experiments.